# Tailoring superstructure units for improved oxygen redox activity in Li-rich layered oxide battery's positive electrodes

Hao Liu[1], Weibo Hua [1,2] ✉, Sylvia Kunz[3], Matteo Bianchini [3], Hang Li[1,4], Jiali Peng[1], Jing Lin[5], Oleksandr Dolotko[1], Thomas Bergfeldt [1], Kai Wang[5,6], Christian Kübel [5,6,7,8], Peter Nagel[8,9], Stefan Schuppler[8,9], Michael Merz [8,9], Bixian Ying[10], Karin Kleiner [10], Stefan Mangold[11], Deniz Wong [12], Volodymyr Baran[13], Michael Knapp [1], Helmut Ehrenberg [1] & Sylvio Indris [1,14] ✉

The high-voltage oxygen redox activity of Li-rich layered oxides enables additional capacity beyond conventional transition metal (TM) redox contributions and drives the development of positive electrode active materials in secondary Li-based batteries. However, Li-rich layered oxides often face voltage decay during battery operation. In particular, although Li-rich positive electrode active materials with a high nickel content demonstrate improved voltage stability, they suffer from poor discharge capacity. Here, via physico-chemical and electrochemical measurements, we investigate the correlation between oxygen redox activity and superstructure units in Li-rich layered oxides, specifically the fractions of $LiMn_6$ and $Ni^{4+}$-stabilized $LiNiMn_5$ within the TM layer. We prove that an excess of $LiNiMn_5$ hinders the extraction/insertion of lithium ions during Li metal coin cell charging/discharging, resulting in incomplete oxygen redox activity at a cell potential of about 3.3 V. We also demonstrate that lithium content adjustment could be a beneficial approach to tailor the superstructure units. Indeed, we report an improved oxygen redox reversibility for an optimized Li-rich layered oxide with fewer $LiNiMn_5$ units.

In contrast to conventional layered positive electrode oxides, such as $LiCoO_2$, relying solely on transition metal (TM) redox activity, Li-rich layered oxides have emerged as promising positive electrode materials due to their utilization of both TM and oxygen redox at high voltage, resulting in an improved discharge capacity[1]. However, the instability of oxygen redox triggers severe voltage decay, posing a substantial challenge for the implementation of commercial applications[2,3]. Over the past decades, intensive studies have been proposed to elucidate the origins of voltage decay, including the continuous reduction of TM valence states, irreversible TM ions migration, and phase transitions (from layered to spinel/rock-salt phases) during cycling[4-7]. Recently, a

more profound understanding[8] has emerged, highlighting the crucial role of the accumulation of mesoscale lattice strain and lattice displacement within distinct nanoscale domains ($LiTMO_2$ and $Li_2MnO_3$) as the driving force behind voltage decay during battery operation. Consequently, conventional post-synthesis treatments such as surface coating have proven ineffective in addressing this issue.

Tailoring the local structure and optimizing chemical composition is emerging as a promising avenue to mitigate voltage decay in Li-rich layered oxides. For instance, a consensus has emerged from relevant studies[9-11], highlighting the detrimental role of localized superstructure domains characterized by in-plane Li/Mn order of Li-

rich oxides, while delocalized or dispersed domains characterized by in-plane Li/Mn disorder effectively improve the oxygen redox reversibility and voltage stability. In addition, Li-rich oxides with O2-type exhibit improved voltage stability compared to conventional O3-type oxides, attributed to the reversible migration of TM ions[12,13]. In particular, an O2-type Li-rich oxide with a capped-honeycomb structure demonstrates negligible voltage decay[14]. However, synthesizing these positive electrodes through an ion exchange method from P2-type sodium positive electrode precursors presents challenges for large-scale production[15]. Introducing concentration gradients in Li-rich layered oxides, with low manganese and high nickel content on the particle surface, has shown improved voltage retention[16,17]. Nevertheless, uncertainties in nickel/manganese ions migration during high-temperature calcination lead to deviations from the intended design, compromising materials synthesis reproducibility. In terms of optimizing chemical composition, high-nickel Li-rich layered oxides enhance voltage stability by promoting the formation of $Ni^{3+}$ in the pristine material. This serves as a redox buffer, suppressing the redox activation of $Mn^{4+/3+}$ at low potentials[18,19]. Recently, Li et al.[20] demonstrated the capability of nickel over cobalt in slowing the kinetics of ligand-to-metal charge transfer, mitigating TM ions migration, and resulting in reduced oxygen release and lower voltage decay. Moreover, Li-rich layered oxides can be obtained through hydroxide/carbonate co-precipitation, which is currently the preferred synthesis method for large-scale industrial production[21]. Despite their potential, high-nickel Li-rich layered oxides, like $Li_{1.2}Ni_{0.4}Mn_{0.4}O_2$, have received less attention due to their unsatisfactory electrochemistry, including lower discharge capacity and slower lithium ions diffusion compared to low-nickel Li-rich layered oxides[22–25]. However, there is still no comprehensive explanation for the unsatisfactory electrochemistry of high-nickel Li-rich layered oxides. Thus, the key to advancing Li-rich layered oxides is achieving voltage stability in high-nickel systems without compromising discharge capacity.

In this work, we present a comprehensive investigation of cobalt-free Li-rich layered oxides with varying nickel contents ($Li_{1.20}Ni_xMn_{0.8-x}O_2$, $x = 0.28, 0.32, 0.36, 0.40$). By employing advanced techniques including synchrotron X-ray diffraction (SXRD), X-ray pair distribution function (PDF) analysis, X-ray absorption spectroscopy (XAS), solid-state nuclear magnetic resonance (NMR) spectroscopy, and density functional theory (DFT) calculations, we identify the types of honeycomb superstructure units within the TM layers and elucidate the influence of the valence state of nickel ions on these units. Furthermore, by integrating electrochemical investigations with soft XAS and resonant inelastic X-ray scattering (RIXS) spectroscopy, we establish a direct correlation between oxygen redox behaviors and the ratio of superstructure units. This correlation provides insights into the underlying reasons for the poor electrochemistry observed in high-nickel Li-rich layered oxides. Finally, we propose a practical solution to achieve improved electrochemical performance with reversible oxygen redox by tailoring honeycomb superstructure units in high-nickel Li-rich layered oxides.

## Results

### Structural characterization and electrochemistry
The $Li_{1.20}Ni_xMn_{0.8-x}O_2$ materials with $x$ values of 0.28, 0.32, 0.36, and 0.40, denoted as N28, N32, N36, and N40, respectively, were synthesized via a solid-state reaction involving TM hydroxide precursors and $Li_2CO_3$ (with a molar ratio of Li/TM = 1.5) under ambient air conditions (see "Methods"). Scanning electron microscopy (SEM) images (Supplementary Fig. 1) show that all materials have a typical polycrystalline microstructure with spherical secondary particles, each approximately 6 μm in diameter. High-resolution synchrotron X-ray diffraction (SXRD) measurements were used to obtain the crystal structure information. As shown in Supplementary Fig. 2, the main Bragg reflections for N28, N32, N36, and N40 correspond to a layered

rhombohedral (R-3m) phase without impurities. Additionally, the broad diffraction peaks at $2\theta$ values of 2.7–3.1° indicate honeycomb Li/TM ordering (C2/m space group) within the TM layer[8]. Meanwhile, these superstructure diffraction peaks show similar intensities (see enlarged image on the right), indicating all materials have comparable amounts and faulting of honeycomb domains. To obtain detailed structural parameters, we conducted SXRD refinements (Fig. 1a). For simplicity, we used a single R-3m phase as the initial structural model. The detailed refinement results are shown in Supplementary Table 1. The value of Li/Ni disorder is 2.4(4), 2.3(4), 2.2(4), and 1.8(4)% for N28, N32, N36, and N40, respectively, indicating that the cation arrangements in these materials are similar. To capture local structural arrangement, X-ray pair distribution function (PDF) measurements were conducted. The PDF patterns of $LiTMO_2$ and $Li_2MnO_3$ are calculated and shown in Supplementary Fig. 3. The first two peaks below 3.0 Å represent the octahedral TM-O and the nearest TM-TM atom pairs, respectively[26]. The intensity of the second PDF peak ($\sim 2.9$ Å) differs significantly between the R-3m model (mostly TM-TM pairs) and the C2/m model (partial TM-TM and partial Li-TM), leading to a distinct difference in the intensity ratio of the first and second PDF peaks. This distinction arises from the unique Li/Mn ordering in $Li_2MnO_3$. Consequently, the intensity of the second PDF peak is significantly weakened in the C2/m due to the relatively poor X-ray scattering capability of the Li atoms. As shown in Fig. 1b, the PDF patterns of N28, N32, N36, and N40 show no noticeable differences in the first two PDF peaks, indicating nearly identical atomic arrangements across all materials. High-resolution transmission electron microscopy (HRTEM) measurements were performed to verify local atomic arrangements. The HRTEM images (Supplementary Fig. 4) of N28, N32, N36, and N40 show clear lattice fringes without obvious evidence of Li/Ni mixing, consistent with the SXRD refinements. Additionally, the observed average inter-planar spacing of approximately 0.47 nm corresponds to the $(003)_R$ or $(001)_M$ plane of the layered rhombohedral (R-3m) or monoclinic (C2/m) structures.

We then evaluated the electrochemical performance of these materials using Li metal coin cells with non-aqueous liquid electrolyte solution at a rate of 20 mA g$^{-1}$ within the voltage range of 2.0–4.7 V at 25 °C. Figure 1c displays the characteristic voltage profiles of the Li metal cells with Li-rich layered oxides at the positive electrode, featuring a sloping part below 4.45 V and a subsequent plateau above 4.45 V[8]. The former is associated with cationic redox activity in $LiTMO_2$ domains, whereas the latter involves anionic (oxygen) redox activity in $Li_2MnO_3$ domains. Despite comparable total charge capacities, a notable divergence emerges in discharge capacity with increasing nickel content, leading to a gradual decrease. For example, Li||N28 coin cell and Li||N40 coin cell enable discharge capacities of 235.6 and 168.8 mAh g$^{-1}$, with initial columbic efficiencies (ICE) of 71.4% and 58.1%, respectively. During discharging, all coin cells display nearly overlapping curves above 3.7 V. The primary difference arises from the reduction plateau at approximately applied potential of 3.3 V, which gradually disappears, as shown in the differential capacity (dQ/dV) curves depicted in Fig. 1d (indicated by the green arrow). Typically, the reduction peaks around 4.4 and 3.7 V are associated with the reduction of oxygen and nickel at high and mid potentials, respectively. The peak below 3.7 V, corresponding to low potential, probably relates to the reduction of oxygen or manganese[1]. Distinguishing these peaks is challenging in low-nickel Li-rich oxides due to their overlap. However, in high-nickel Li-rich oxides, the low-potential peak nearly disappears as nickel content increases. Thus, high-nickel oxides serve as an ideal model for investigating the redox mechanism in this region.

### Identification of incomplete oxygen redox activity
To verify the redox nature of the dQ/dV peak observed in the region below 3.7 V, we conducted soft X-ray absorption spectroscopy (SXAS) measurements, using N36 as an example. Ni and O were analyzed in the

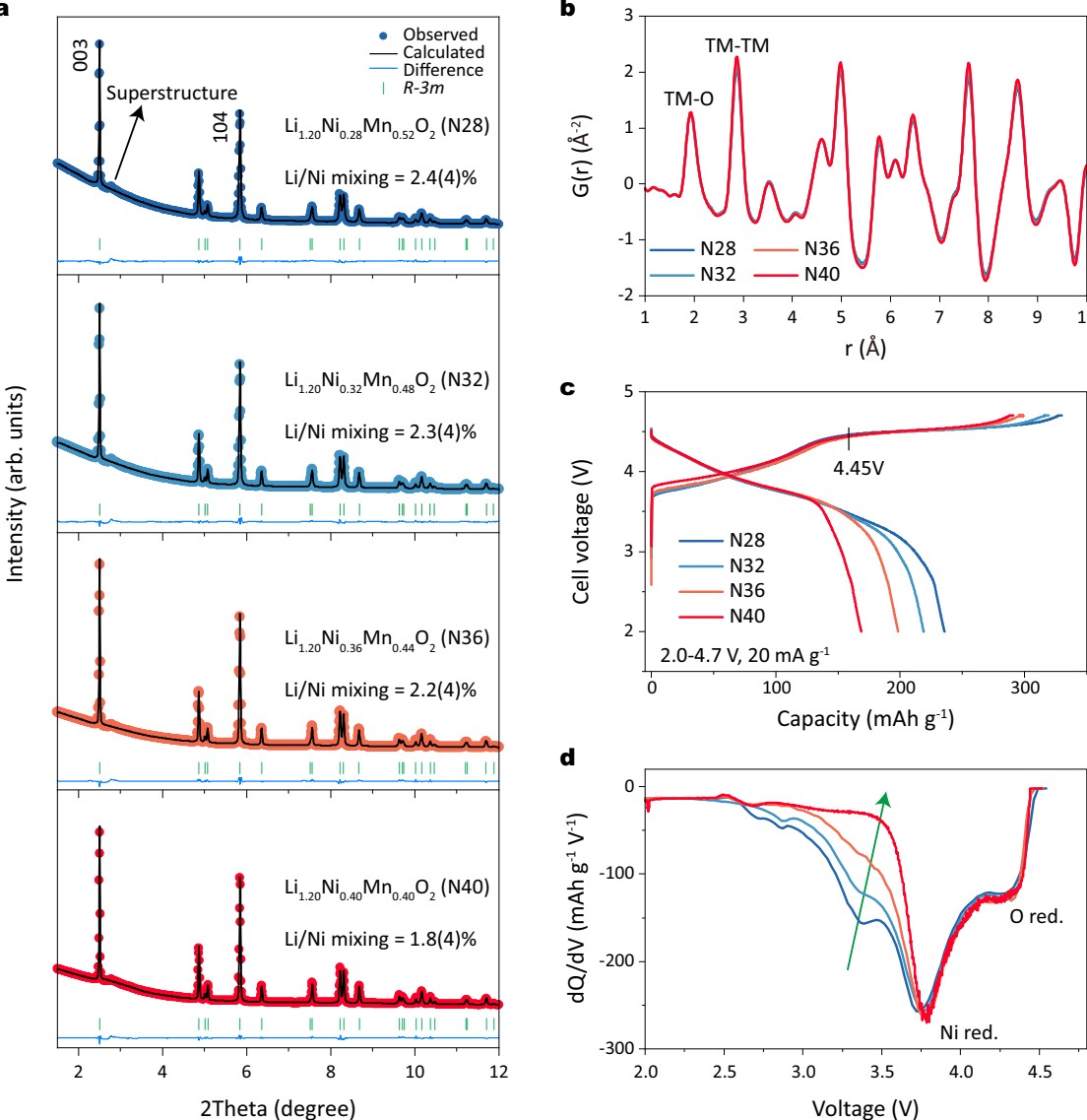

**Fig. 1 | Structural characterization and electrochemistry. a** SXRD patterns and corresponding refinement results for N28, N32, N36 and N40 powders. The Rietveld refinement was performed with space group *R-3m*. The wavelength at the beamline was 0.2073 Å. **b** PDF patterns of N28, N32, N36 and N40 powders.

**c** Charge and discharge curves and (**d**) corresponding dQ/dV curves of Li metal coin cell with the N28, N32, N36, and N40-based positive electrodes in the first cycle in the voltage range of 2.0–4.7 V at 20 mA g⁻¹ at 25 °C.

fluorescence-yield mode (FY), while Mn was examined in the inverse partial-fluorescence-yield mode (iFY) at different charge/discharge states (as marked in Fig. 2a). FY and iFY measurements probe to a depth of approximately 100 nm, thus providing information about the bulk of the material. Charging to 4.45 V led to Ni oxidation, evidenced by a decrease in peak A intensity ( ~ 852.9 eV) and an increase in peak B ( ~ 854.8 eV) in the Ni L3-edge spectra (Fig. 2b)[27]. At the end of the 4.7 V charge, the spectra exhibited slightly inverted changes, possibly related to charge transfer from ligand to metal[20]. Upon discharging to 2.0 V, reversed changes in the Ni L3-edge spectra indicated Ni reduction, with peak A showing higher intensity and peak B lower intensity compared to the pristine state, suggesting a reduction of nickel species to a lower valence state. The Mn L-edge spectra remained constant, indicating Mn stability in an inactive $Mn^{4+}$ valence state throughout the entire process (Fig. 2c).

In the O K-edge spectra (Fig. 2d), the pre-edge region (below 535 eV) primarily reflects the electron transition from O 1s level to the unoccupied TM 3d-O 2p hybridized states[28]. Charging to 4.45 V induces increased intensity at around 528.5 eV (marked as a gray

arrow), attributed to Ni oxidation, consistent with prior analysis. Additionally, a new peak emerges at approximately 530.8 eV (dashed lines), becoming more pronounced upon charging to 4.7 V. This peak is recognized as characteristic of oxidized oxygen species, as observed in the Li-rich layered oxides[29,30]. Notably, this peak associated with oxidized oxygen retains some intensity after discharging of 2.0 V, failing to fully revert to its pristine state, suggesting incomplete oxygen redox reaction within the material. To further elucidate the nature of oxidized oxygen, we conducted O K-edge resonant inelastic X-ray scattering (RIXS) measurements. In Fig. 2e, at 4.7 V in the charged state, N36 exhibits a distinct characteristic peak C (around 7.8 eV, denoted by the dashed line) and vibrational peak D around the elastic region (0 eV). In addition, the enlarged view of the vibrational peak is presented in Fig. 2f, revealing vibrational frequencies of 1498 cm⁻¹, similar to the O-O bond length of molecular $O_2$, aligning with recent research findings[31,32]. More importantly, after discharging to 2.0 V, the distinctive feature of molecular $O_2$ persists (indicated by the gray arrow), suggesting an incomplete reduction of molecular $O_2$ to $O^{2-}$

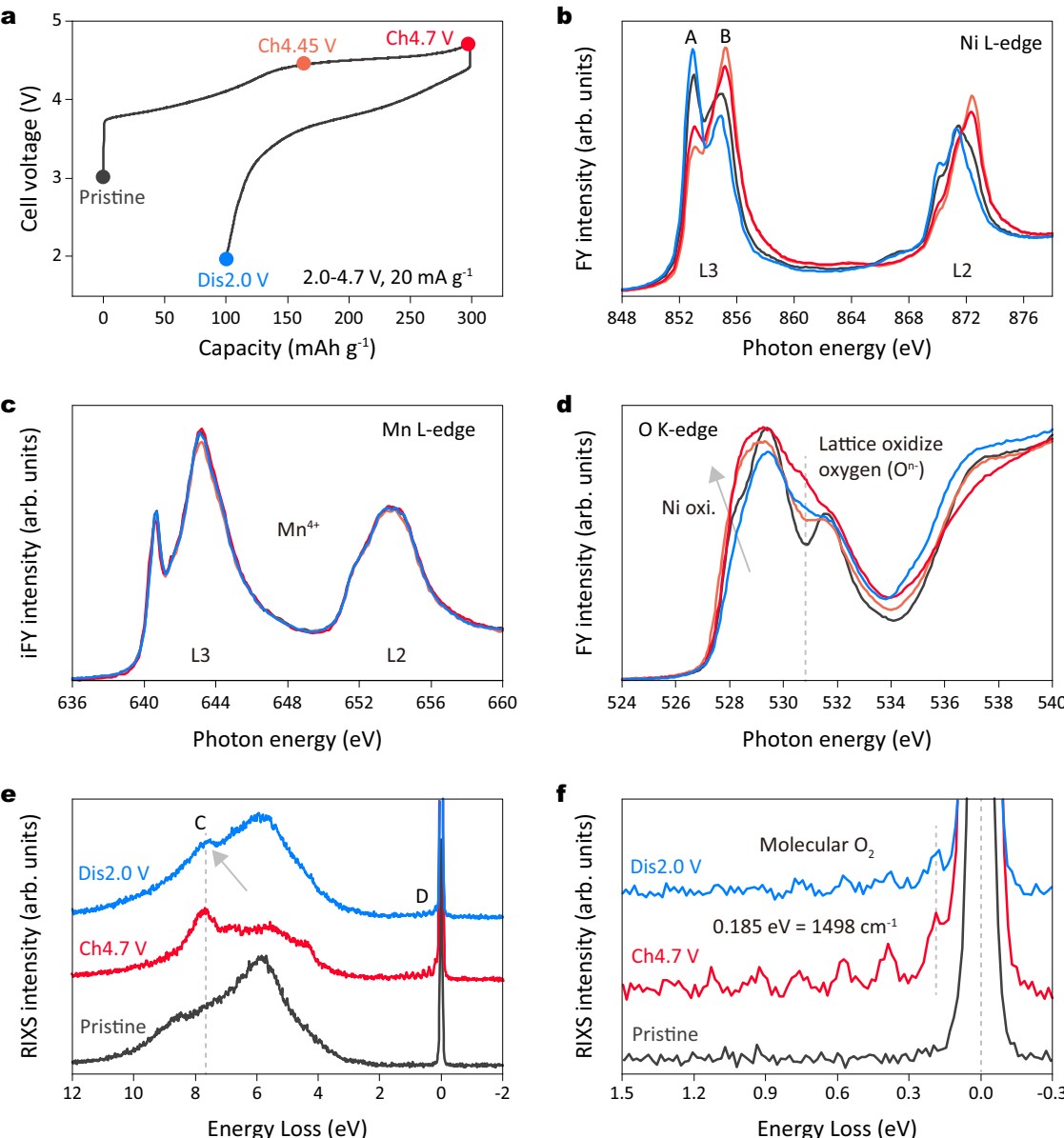

**Fig. 2 | Identification of incomplete oxygen redox activity. a** Initial charge and discharge curves of Li||N36 coin cell in the voltage range of 2.0-4.7 V at 20 mA g$^{-1}$ with specific points marked for ex-situ SXAS measurements. SXAS results of the (**b**) Ni L-edge, (**c**) Mn L-edge, and (**d**) O K-edge collected at pristine, charge to 4.45 V, charge to 4.7 V, and discharge to 2.0 V states. The FY mode was applied for Ni and O and the iFY mode was applied for Mn. **e** O K-edge RIXS spectra collected at an excitation energy of 531 eV at pristine, charge to 4.7 V, and discharged to 2.0 V states in the first cycle and (**f**) an enlarged view of the elastic region (0–1 eV).

species. These findings are consistent with the SXAS results. Thus, we identify that the incomplete oxygen redox occurs at approximately applied potential of 3.3 V in Li-rich layered oxides for a high-nickel system.

**Superstructure characterization and theoretical calculation**
Typically, molecular O$_2$, in the charged state, is trapped in vacancy clusters within the TM layer of Li-rich layered oxides[31]. Our experiments have demonstrated the presence of molecular O$_2$ at the end of discharge in high-nickel Li-rich layered oxides. This suggests that lithium ions cannot smoothly re-enter the TM layer to coordinate with oxygen during discharge, leaving some oxygen species as molecular O$_2$. Therefore, the issue may lie in the TM layer, specifically at the local honeycomb superstructure. To detect the honeycomb superstructure distribution, we conducted $^6$Li solid-state nuclear magnetic resonance (NMR) spectroscopy measurements. Figure 3a and Supplementary Table 2 present the $^6$Li NMR patterns with fitting results. The sharp

peak at 0 ppm represents diamagnetic species such as LiOH, Li$_2$CO$_3$, and organic lithium salts[33]. Two prominent groups of peaks are observed at 500–1000 ppm and 1300–1500 ppm, corresponding to Li in the Li layer (Li$_{Li}$) and Li in the TM layer (Li$_{TM}$), respectively[34]. The NMR patterns reveal two distinct peaks at approximately 1300 and 1500 ppm, indicating two different Li environments within the TM layer. Consistent with previous literature[35], these peaks are assigned to honeycomb superstructure units for LiMn$_6$ (purple, Li surrounded by six Mn) and LiNiMn$_5$ (blue, Li surrounded by one Ni and five Mn), respectively. The corresponding atomic structures are plotted in Fig. 3b. Interestingly, although all materials show similar total lithium content in the TM layer, the local lithium environments differ. Specifically, NMR patterns reveal a gradual increase in the intensity of peaks corresponding to LiNiMn$_5$ units, while the intensity of peaks for LiMn$_6$ units decreases from N28 to N40. Fitting results show that the ratio of LiNiMn$_5$/(LiNiMn$_5$ + LiMn$_6$) increases progressively: 0.183 for N28, 0.373 for N32, 0.540 for N36, and 0.613 for N40 (Fig. 3c). Thus, the

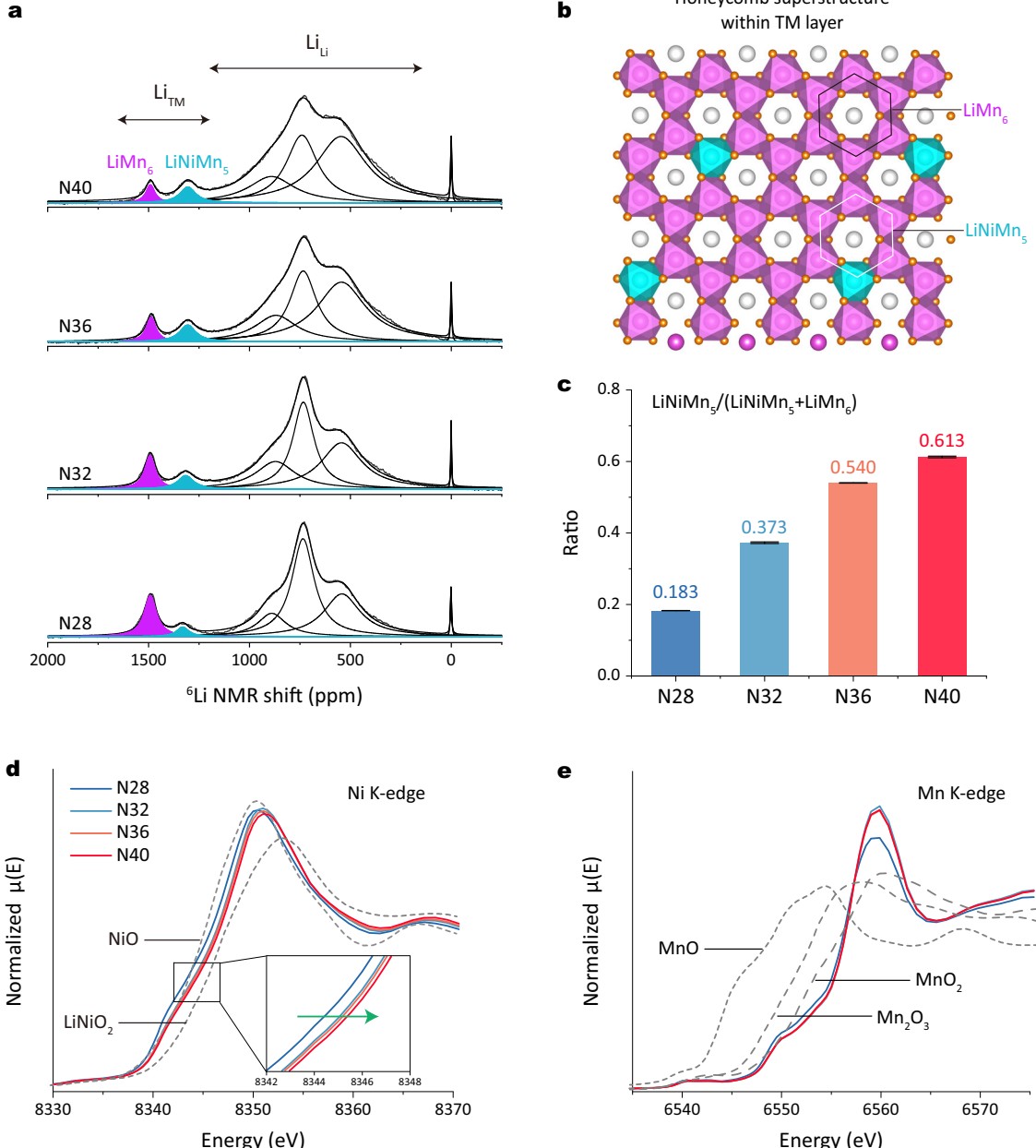

**Fig. 3 | Superstructure units characterization. a** $^6$Li NMR spectra and fitting results of N28, N32, N36, and N40 powders. NMR fitting was performed with the DMFIT program. **b** In-plane ordering of Li (gray) and Mn (purple octahedra) or Ni (blue octahedra) within TM layers forming LiMn$_6$ and LiNiMn$_5$ superstructure units.

**c** Ratio of LiNiMN$_5$/(LiNiMn$_5$ + LiMn$_6$). The error bars in **c** correspond to the s.d. of three independent fitting procedures. Data are given as average ± s.d. Normalized XAS (**d**) Ni and (**e**) Mn K-edge XANES of N28, N32, N36, and N40 electrodes. The XAS data are collected in transmission mode.

varying Ni/Mn ratios in Li-rich layered oxides affect the type of superstructure units within the TM layer. As nickel content increases, the Li-rich oxide exhibits more LiNiMn$_5$ units, replacing LiMn$_6$ units.

Since it is difficult to detect LiNiMn$_5$ superstructure units in conventional Li-rich layered oxides (e.g. Li$_{1.2}$Ni$_{0.2}$Mn$_{0.6}$O$_2$), the potential impact of this unit on electrochemistry may have been overlooked. To understand the formation mechanism of LiNiMn$_5$, we conducted density functional theory (DFT) calculations. We chose Li$_{44}$Mn$_{18}$Ni$_{10}$O$_{72}$ (Li$_{1.22}$Mn$_{0.50}$Ni$_{0.28}$O$_2$, LiNiMn$_5$ within the TM layer) as the structural model (Supplementary Fig. 5a). By calculating the spin density of all atoms, we confirmed their valence states. The calculations show Mn with +4 valence state, Ni with +2, +3, and +4 valence states, and O with -2 valence state. Importantly, Ni occurs exclusively as Ni$^{4+}$ within the highly symmetric LiNiMn$_5$ units, whereas Ni$^{2+}$ or Ni$^{3+}$ are

identified in more Ni-rich local configurations. Note that we have calculated the oxidation state of Ni in the LiNiMn$_5$ configuration without a priori assumptions on the Ni oxidation state. The relaxed structure shows that Ni possesses the oxidation state +4. In addition, we have calculated models with the same overall composition but without the LiNiMn$_5$ configuration (always without assuming the oxidation state a priori), in which we find that Ni has an oxidation state lower than +4. These findings are supported by the density of states (DOS) analysis (Supplementary Fig. 5b). The detailed calculation processes are described in the "Methods" section. To confirm the Ni valence state changes in these materials, we conducted XAS measurements. As shown in Fig. 3d, the Ni K-edge spectra shift to higher energy from N28 to N40, indicating an increase in the average Ni oxidation state. In contrast, the Mn K-edge spectra position remains constant for all

materials, indicating an $Mn^{4+}$ oxidation state (Fig. 3e). These valence state changes are further supported by the Fourier transform extended X-ray absorption fine structures (FT-EXAFS) results, shown in Supplementary Fig. 6. While $Ni^{4+}$ is atypical in conventional layered oxides, its presence in Li-rich layered oxide systems is plausible due to the lithium excess structural configuration[36]. Consequently, forming $LiNiMn_5$ superstructure units is easier in N40 than in N28, as the higher Ni valence state in N40 allows for more $Ni^{4+}$ to be present.

In addition, Pauling's electrostatic valence principle[37] and semi-quantitative equation[38] can be used to evaluate the possibility of a coordination configuration. According to principle of local electrical neutrality, a structure may become unstable if $\triangle Z > 0$. The most common $LiMn_6$ superstructure units, equivalent to $Li_3$-O-$LiMn_2$, exhibit a stable state with $\triangle Z = 0$. Similarly, $LiNiMn_5$ units ($Li_3$-O-$LiNiMn$) also demonstrate $\triangle Z = 0$, attributed to the presence of $Ni^{4+}$ rather than $Ni^{2+}$ ($\triangle Z = 1/3$) and $Ni^{3+}$ ($\triangle Z = 1/6$). This aspect elucidates why prior investigations failed to detect nickel within $Li_2MnO_3$-like domains, given that the possibility of the presence of $Ni^{4+}$ in $LiNiMn_5$ superstructure units has been previously overlooked[38,39]. Previous theoretical calculations[39] indicate that the $Li_4MnNi$-coordinated O, referred to as $LiNiMn_5$ in our work, exhibits lower cationic and anionic redox activity compared to $LiMn_6$ units, indicating limited electrochemical activity. Moreover, lithium ions move from one octahedral site to another, passing through an intermediate tetrahedral site where they encounter strong repulsion from nearby TM ions. The associated activation barriers for lithium movement with $Ni^{4+}$, $Ni^{3+}$ and $Ni^{2+}$ are 490, 310, and 210 meV, respectively[40]. These theoretical findings imply that $LiNiMn_5$ may impede the extraction/insertion of lithium ions. During charging/discharging, Li-rich oxides experience a cationic–anionic redox inversion, leading to electrochemical asymmetry[27]. As a consequence, at low discharge potentials, some lithium ions fail to return to the TM layer to coordinate with oxygen, resulting in incomplete oxygen redox. This analysis is consistent with the findings from O K-edge SXAS and RIXS experiments.

Combining all the analysis described above, a clear correlation between the incomplete oxygen redox reaction and honeycomb superstructure units can be established. In Li-rich layered oxides, as the nickel valence state increases (reflecting increased nickel content from N28 to N40), positive electrodes exhibit a rise in $LiNiMn_5$ at the expense of $LiMn_6$ units. Positive electrodes with higher $LiNiMn_5$ content exhibit limited electrochemical activity, resulting in incomplete oxygen redox reactions (retention of molecular $O_2$) at a low potential of 3.3 V, thereby leading to decreased discharge capacity. In contrast, Li-rich oxides with less $LiNiMn_5$, such as N28, demonstrate enhanced oxygen redox reversibility, showing higher discharge capacity.

### Reversible oxygen redox activity through tailoring superstructure units

To address these intrinsic properties in high-nickel Li-rich layered oxides, we propose a practical solution: reducing the lithium content to lower the Ni valence state and thus tailor the $LiNiMn_5$ superstructural units. During the synthesis, we varied the lithium content and the ratio of TM precursors. Specifically, we used Li/TM ratios of 1.2, 1.3, 1.4, and 1.5 based on the N36 precursor $Ni_{0.45}Mn_{0.55}(OH)_2$. The resulting chemical formulas are $Li_{1.09}Ni_{0.41}Mn_{0.50}O_2$ (LM12), $Li_{1.13}Ni_{0.39}Mn_{0.48}O_2$ (LM13), $Li_{1.17}Ni_{0.37}Mn_{0.46}O_2$ (LM14), and $Li_{1.20}Ni_{0.36}Mn_{0.44}O_2$ (LM15). It is important to note that LM15 is equivalent to the N36 mentioned earlier. Inductively coupled plasma-optical emission spectroscopy (ICP–OES) results confirm that the Li/TM molar ratios are consistent with the design values, except for some lithium evaporation at high temperatures (Supplementary Table 3). As shown in Supplementary Fig. 7, LM12, LM13, LM14, and LM15 maintain a layered phase with space group $R\text{-}3m$ without impurities. In the enlarged view on the right, the diffraction intensity related to the superstructure decreases from LM15 to LM12, indicating a reduction in

superstructure content as lithium content decreases. The results of the SXRD refinement and corresponding data are presented in Supplementary Fig. 8 and Supplementary Table 4. The Li/Ni mixing reveals an increasing level of disorder from LM15 to LM12, with LM15 displaying a minimal disorder value of 1.8(4)%, while LM12 shows a significantly higher disorder value of 6.3(3)%. Considering both discharge capacity and cycle stability, LM13 is selected as the optimized positive electrode for further discussion (Supplementary Fig. 9).

The PDF patterns of both cathodes are shown in Supplementary Fig. 10. LM13 shows a higher intensity of the second PDF peaks correspond to TM-TM pairs compared to LM15 (indicated by the arrow). It suggests LM13 exhibits more $R\text{-}3m$ than $C2/m$ domains, which agrees with SXRD analysis. Additionally, both materials exhibit a uniform distribution of Ni, Mn, and O, confirmed by transmission electron microscopy (TEM) energy dispersive X-ray spectroscopy (EDS) mapping images (Supplementary Fig. 11). To investigate the distribution of honeycomb superstructure units, we conducted $^6Li$ NMR measurements of LM13. Compared to LM15, LM13 exhibits a lower NMR peak intensity at approximately 1300 ppm, indicating fewer $LiNiMn_5$ units (Fig. 4a). The NMR fitting results (Supplementary Table 5) show that the ratio of $LiNiMn_5/(LiNiMn_5 + LiMn_6)$ is 0.427 for LM13, lower than 0.540 for LM15 (Fig. 3c). Meanwhile, XAS results show a lower Ni oxidation state in LM13, while Mn maintains the same oxidation state ( + 4) for both materials (Fig. 4b and Supplementary Fig. 12). The XAS analysis is further supported by the corresponding FT-EXAFS results, shown in Supplementary Fig. 13. Overall, through our comprehensive analyzes (including SXRD, PDF, XAS, and NMR), we have successfully tuned the composition and content of the honeycomb superstructure units simultaneously. In Fig. 4c, Li||LM13 coin cell exhibits a longer sloping part (below 4.45 V) and a shorter oxygen redox plateau than Li||LM15 coin cell, reflecting differences in superstructure contents. Despite a slight trade-off in charge capacity due to lower lithium content, Li metal coin cell with the LM13-based positive electrode demonstrates a notable capacity of 231.1 mAh $g^{-1}$ with an ICE of 78.4%. This surpasses the 199.0 mAh $g^{-1}$ of Li||LM15 coin cell (ICE of 69.8%). The dQ/dV curves in Fig. 4d highlight the significant difference in discharge capacity, attributed to the co-contribution of nickel redox and oxygen redox (especially at a low potential of 3.3 V, indicated by the green arrow). This modulation of cationic and anionic redox aligns precisely with the impact of superstructure content and units in Li-rich layered oxides. To confirm the reversibility of oxygen redox in the LM13, we conducted O K-edge RIXS measurements. Figure 4e shows a clear molecular $O_2$ signature during charging, indicating oxygen oxidation. In contrast to LM15 (Fig. 2e), no signal is detected at the end of discharge for LM13. The enlarged view of the elastic region (0–1 eV) in Fig. 4f further supports these observations, indicating complete oxygen redox after tailoring the superstructure units. Note that the relatively lower oxygen redox signal in LM13 is attributed to the reduced superstructure content, consistent with above structural analysis.

### Structural evolution analysis

To track the evolution of lithium local environments during charge/discharge, ex-situ $^7Li$ NMR spectroscopy was performed. In Fig. 5a-b, the $^7Li$ NMR results in the pristine state, in agreement with the $^6Li$ NMR measurements described above, confirm fewer $LiNiMn_5$ units in LM13. In the charged state at 4.35 V for LM15, the intensity of the peak corresponding to $LiNiMn_5$ remains unchanged (see green dashed line), indicating minimal changes in the oxidation state of TM ions within $LiNiMn_5$, confirming the presence of $Ni^{4+}$-stabilized $LiNiMn_5$ units rather than $Ni^{2+}$ or $Ni^{3+}$. Beyond 4.5 V, the NMR peak broadens due to structural disorder induced by TM ions migration in Li environments, making it challenging to identify specific superstructure units in both positive electrodes. In the charged state at 4.7 V, some NMR signals persist in the 1300–1500 ppm region in LM15, indicating lithium ions in the TM layers. In sharp contrast, LM13 shows no signal in this region,

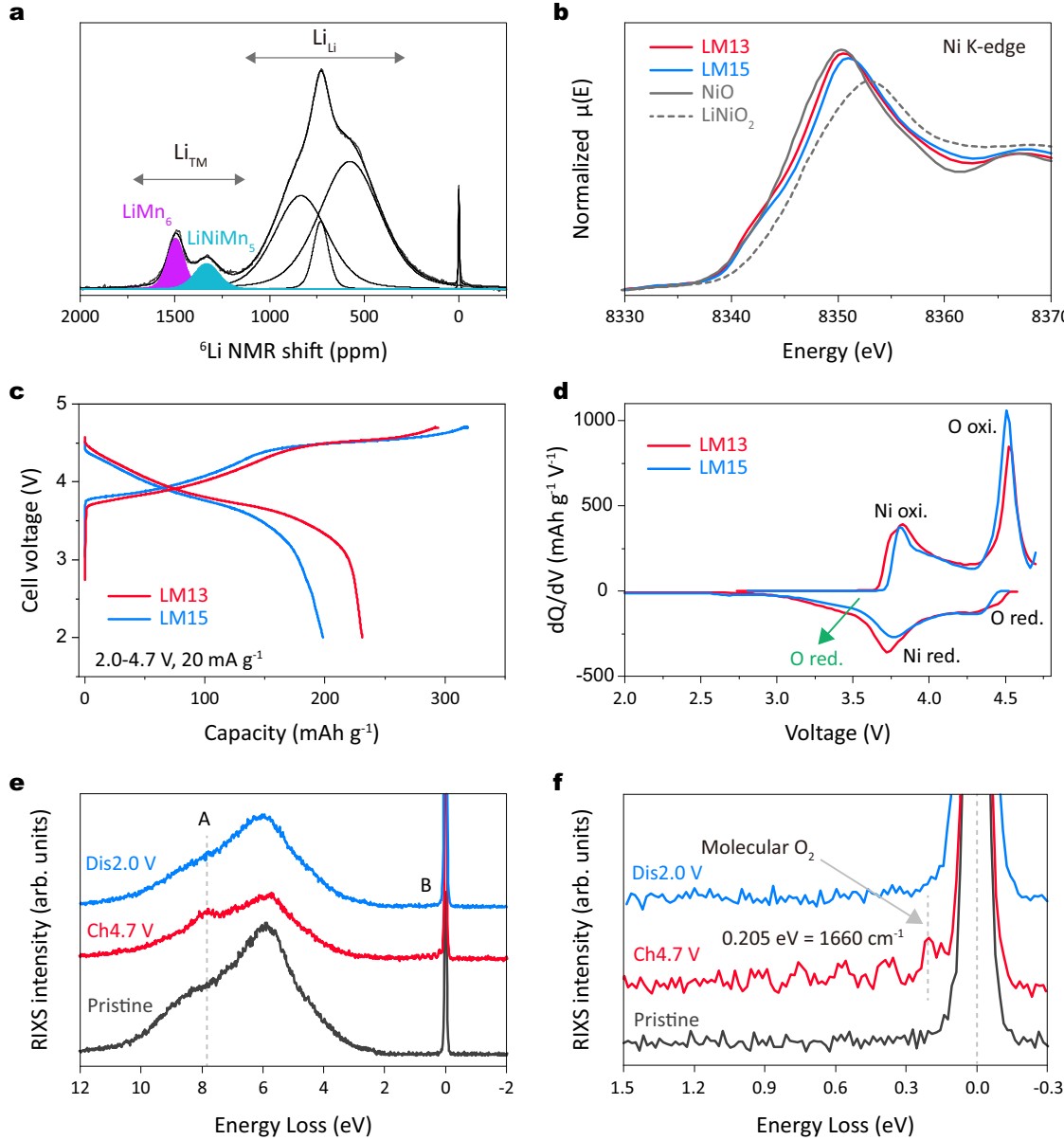

**Fig. 4 | Reversible oxygen redox activity through tailoring superstructure units. a** $^6$Li NMR spectra and fitting results for LM13 powder. **b** Normalized XAS Ni K-edge XANES of LM13 and LM15 electrodes in the pristine state. The XAS data are collected in transmission mode. **c** Charge and discharge curves and (**d**) corresponding dQ/dV curves of Li metal coin cell with the LM13 and

LM15-based positive electrodes in the first cycle in the voltage range of 2.0-4.7 V at 20 mA g$^{-1}$ at 25 °C. **e** O K-edge RIXS spectra collected at an excitation energy of 531 eV at pristine, charge to 4.7 V, and discharged to 2.0 V states in the first cycle and (**f**) an enlarged view of the elastic region (0–1 eV) of LM13 electrodes.

implying complete lithium-ion removal from the TM layer. Moreover, the lower Li$_{Li}$ (500–1000 ppm) intensity suggests fewer lithium ions in the lattice for LM13. During discharge, the peak intensity increases as lithium ions are inserted, with LM13 exhibiting improved capability for extraction/insertion of lithium ions, as indicated by the higher total peak intensity in both Li$_{Li}$ and Li$_{TM}$.

To capture the structural evolution during the first cycle, in-situ XRD measurements were conducted, and lattice parameters along with volume changes were obtained through Rietveld refinement (Fig. 5c-d). Below 4.45 V (indicated by the dashed line), both positive electrodes show an increase in lattice parameter $c$, with the 003 Bragg reflection shifting to lower angles. This shift is attributed to increased electrostatic repulsion accompanying the removal of lithium ions from the Li layer[8]. LM15 exhibits minimal changes in lattice parameter $c$ at the oxygen redox plateau, whereas LM13 undergoes significant lattice

shrinkage, indicating lithium ions being removed from the TM layer, consistent with previous reports on Li-rich layered oxides[41]. The minor changes in LM15, supported by NMR analysis, suggest some lithium ions still reside in the lattice, offsetting the lithium removal effect at the end of the charge. During discharge, both positive electrodes show similar changes, initially increasing and then decreasing in lattice $c$, implying the insertion of lithium ions into the TM and Li layers, respectively. The lattice parameter $c$, however, cannot revert to the initial position, showing a difference of 0.065 Å. In stark contrast, LM13 demonstrates minor changes in parameter $c$, with a change of only 0.015 Å after one complete cycle. The lattice parameter $a$, sensitive to the TM-O bond length in the TMO$_2$ slab, serves as indirect evidence of cationic redox[20]. In both positive electrodes, a linear decrease in the slope region and minor changes at the oxygen redox plateau are observed, corresponding to nickel oxidation and oxygen oxidation,

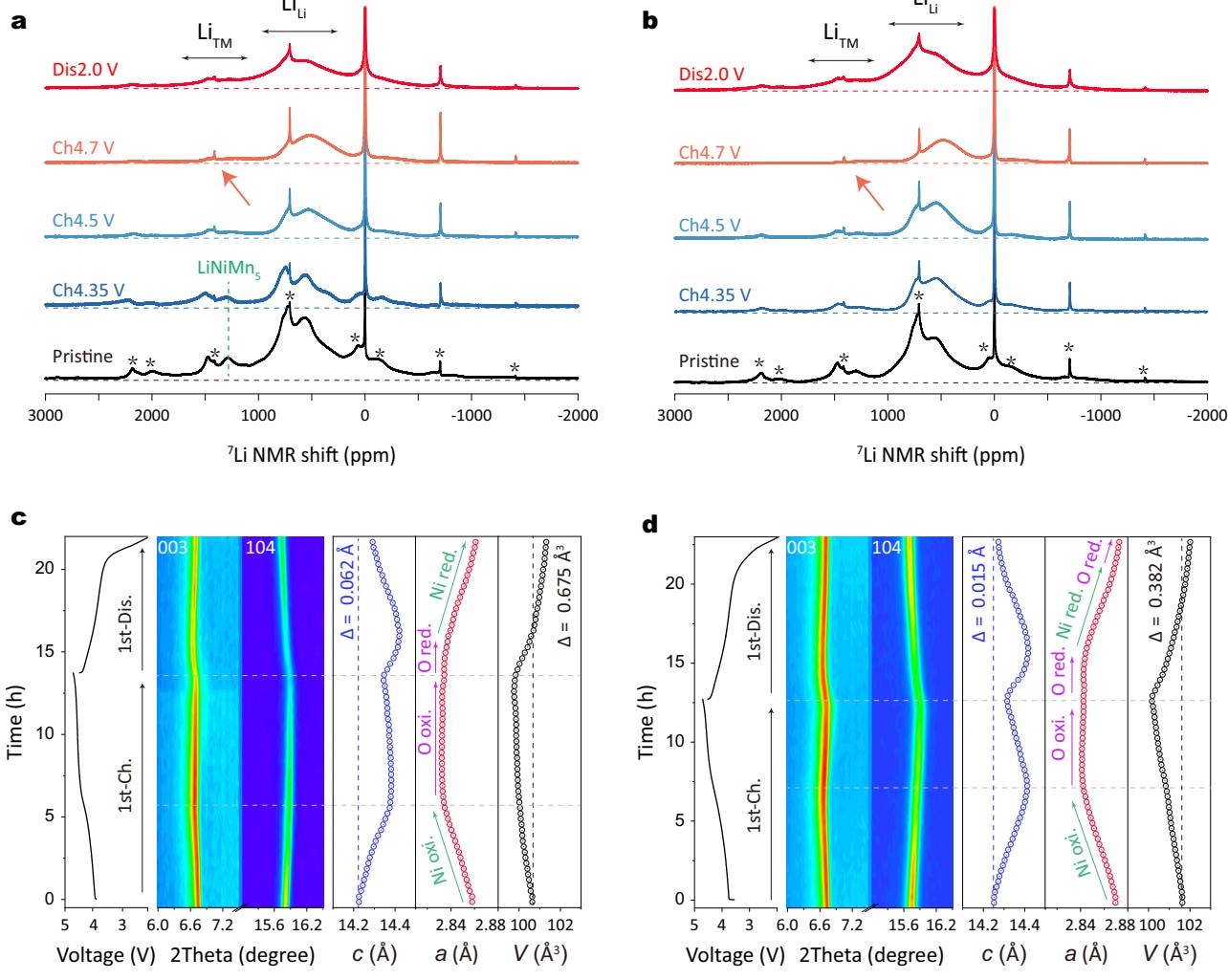

**Fig. 5 | Structural evolution analysis.** $^7$Li MAS NMR spectra of (**a**) LM15 and (**b**) LM13 electrodes in the pristine state and after charging to 4.35, 4.5, and 4.7 V, and discharging to 2.0 V in the first cycle. Spinning sidebands are marked by an asterisk. In-situ XRD of (**c**) Li||LM15 coin cell and (**d**) Li||LM13 coin cell during the first cycle at 20 mA g$^{-1}$ in the voltage range of 2.0-4.7 V at 25 °C. The corresponding lattice parameters and volume were obtained by Rietveld refinement. The in-situ XRD patterns are acquired with an Ag source powder diffractometer in transmission geometry (wavelength of 0.5594 Å).

respectively. During discharge in LM13, lithium ions are inserted into the TM and Li layers, corresponding to oxygen, nickel, and oxygen reduction at high/middle/low potential, respectively (indicated by the colored arrows in Fig. 5d). This redox mechanism evolution aligns with other reports on Li-rich layered oxides[1,42]. However, in LM15, the absence of oxygen reduction at the low potential of 3.3 V, confirmed by O K-edge XAS and RIXS results, means that changes in the lattice parameter *a* correspond to oxygen reduction followed by nickel reduction during discharge. Additionally, both positive electrodes experience volume shrinkage after the first cycle, with LM13 exhibiting a smaller volume change of 0.382 Å$^3$ compared to LM15 (0.675 Å$^3$). This, combined with the previous evolution of the lattice parameter *c*, confirms the superior structural stability of LM13.

**Electrochemistry, chemical and structural stability analysis**
Benefiting from the tailored superstructure content and units, as shown in Fig. 6a, Li||LM13 coin cell displays improved capacity retention at 84.8% (187.7 mAh g$^{-1}$) after 200 cycles at current of 67 mA g$^{-1}$. This contrasts sharply with Li||LM15 coin cell, which exhibits a capacity retention of 71.2% (134.9 mAh g$^{-1}$) based on the maximum capacity. When evaluating Li-rich layered oxides, it is more reasonable to assess the average cell discharge voltage due to the significant

challenge of voltage decay. After 200 cycles, Li||LM13 and Li||LM15 coin cells show average cell discharge voltages of 3.47 and 3.28 V, corresponding to a decrease of 1.36 and 2.43 mV per cycle (Fig. 6b). The selected charge/discharge curves show the distinction in discharge voltage evolution (Supplementary Fig. 14). Furthermore, LM13 demonstrates improved electrochemical energy storage performances at a specific current of 20 mA g$^{-1}$ (Supplementary Fig. 15). Additionally, Fig. 6c highlights the good rate capability of Li||LM13 coin cell, delivering a discharge capacity of 140.9 mAh g$^{-1}$ even at the high rate of 1000 mA g$^{-1}$.

To verify the oxygen redox stability during long-term cycling, we conducted O K-edge RIXS measurements after 100 cycles at 20 mA g$^{-1}$ (approximately 92 days) in the charged state at 4.7 V. As shown in Fig. 6d-e, characteristic features of molecular O$_2$ are clearly observed around an excitation energy of 531 eV and an emission energy of 523.7 eV in LM13 (indicated by the white arrow), while they are barely visible in LM15. Additionally, the O K-edge RIXS spectra collected at an excitation energy of 531 eV (Supplementary Fig. 16) show that LM13 still exhibits a distinct characteristic peak A (around 7.8 eV) and a vibrational peak B around the elastic region (0 eV). In sharp contrast, no such signals are detected in LM15. These results indicate that LM13 has higher oxygen redox stability

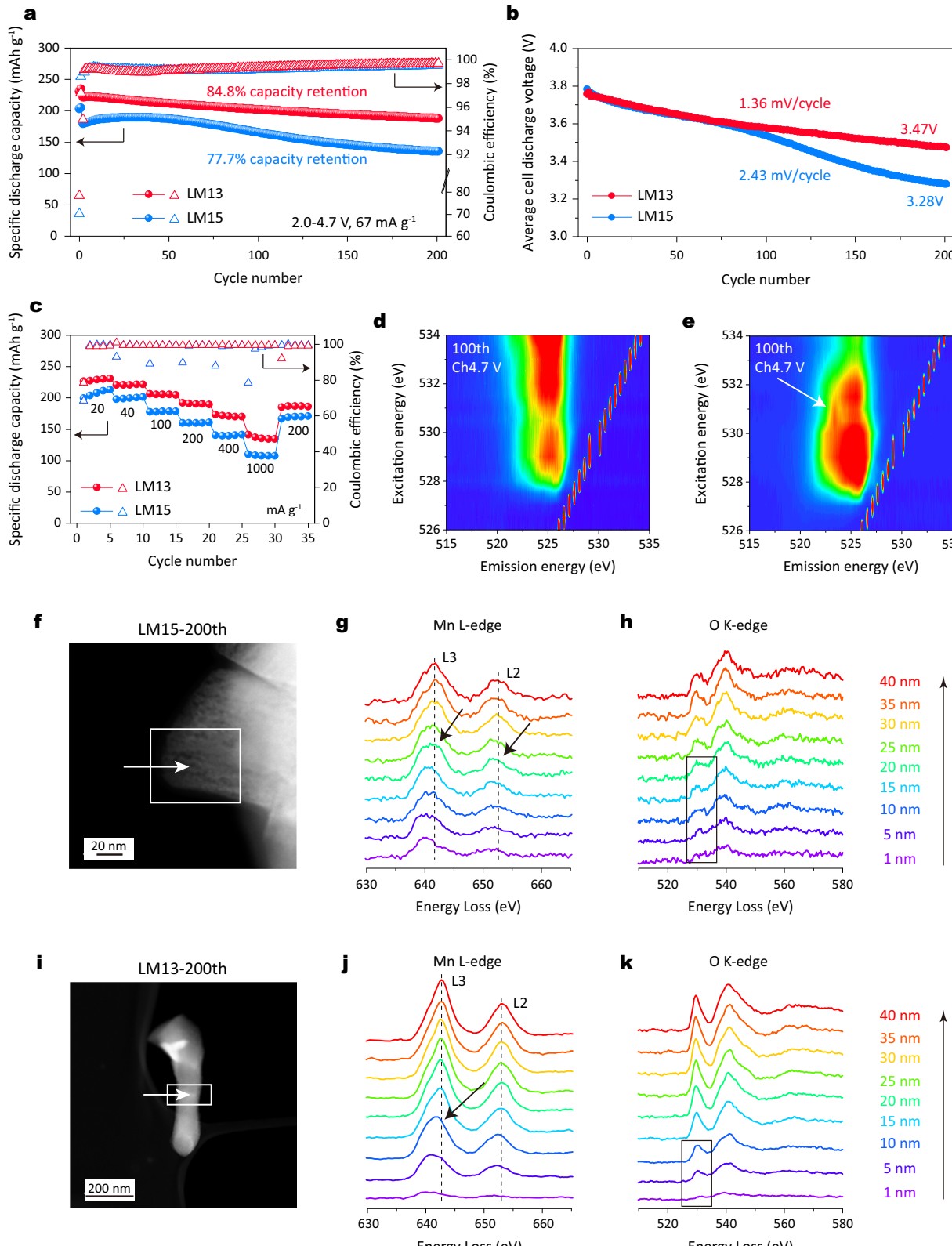

**Fig. 6 | Electrochemistry, chemical and structural stability analysis. a** Cycling performance and (**b**) corresponding average cell discharge voltage curves over 200 cycles at 67 mA g⁻¹ at 25 °C. **c** Rate capability in the voltage range of 2.0-4.7 V at 25 °C. The O K-edge RIXS mapping collected with an excitation energy from 526-534 eV after charging to 4.7 V at 20 mA g⁻¹ after 100 cycles for (**d**) LM15 and (**e**) LM13. EELS line scans at particle surface in the cycled electrode (**f**–**h**) LM15 and (**i**–**k**) LM13 at 67 mA g⁻¹ after 200 cycles.

and reversibility, which explains its improved electrochemical performance. To demonstrate the structural stability of LM13 and LM15, we conducted electron energy loss spectroscopy (EELS) measurements. In the pristine state, as shown in Supplementary Fig. 17, the Mn and Ni L-edge spectra of both electrodes remain at the same energy loss position from the surface to the bulk. However, after 200 cycles, the Mn L-edge spectra of LM15 exhibit a continuous shift to lower energy within a depth of 20 nm (indicated by arrows in Fig. 6g). This shift suggests a gradual reduction of Mn at the surface[43], which is further supported by the fading of the pre-edge peak in the O K-edge spectra (highlighted by the rectangle in Fig. 6h). In contrast, Fig. 6j-k shows that the spectral shifts in LM13 mainly occur up to a depth of 10 nm from the surface. The reproducibility of these results was confirmed by acquiring EELS spectra of the same particle along different directions (Supplementary Fig. 18). Additionally, as shown in Fig. 6f, the TEM images LM15 after 200 cycles clearly reveals extensive voiding in LM15, characterized by a high density of voids distributed throughout the particle. In contrast, LM13 exhibits minimal voids, confirmed by the bright particle images (Fig. 6i). These findings indicate that LM13 demonstrates robust surface stability. To validate the versatility of our strategy for modulating the honeycomb superstructure in high-nickel Li-rich oxides, we applied it to $Li_{1.2}Ni_{0.4}Mn_{0.4}O_2$, maintaining the same Li/TM ratio of 1.3 during synthesis (Supplementary Fig. 19). Similarly, improved electrochemical performance is also observed for the modified oxide (Li/TM = 1.3). Overall, our results confirm that adjusting lithium content is an effective and straightforward approach to enhancing the discharge capacity of high-nickel Li-rich layered oxides.

## Discussion

In summary, using a combination of spectroscopic techniques, DFT calculations, and comprehensive electrochemistry analysis, we have successfully elucidated the origin of the $LiNiMn_5$ honeycomb superstructure units and its influence on the electrochemistry, particularly oxygen redox activity in Li-rich layered oxides. Our findings reveal the presence of two distinct honeycomb superstructure units in the TM layers, namely $LiMn_6$ and $LiNiMn_5$. As the nickel valence state increases (with consistent Mn valence), positive electrodes tend to exhibit a higher proportion of $LiNiMn_5$ at the expense of $LiMn_6$ units. We identified that the $LiNiMn_5$ is exclusively stabilized by $Ni^{4+}$ and not by $Ni^{2+}$ or $Ni^{3+}$. Unlike conventional $LiMn_6$ units, $LiNiMn_5$ impedes the extraction/insertion of lithium ions, resulting in incomplete oxygen redox at a low potential of 3.3 V and retaining molecular $O_2$ in the lattice at the end of discharge. Consequently, positive electrodes with a higher proportion of $LiNiMn_5$ units exhibit reduced discharge capacity. Guided by theoretical insights, we propose a practical approach to inhibit $LiNiMn_5$ superstructure units by reducing lithium content during synthesis, resulting in enhanced electrochemical performance with reversible oxygen redox. The optimized oxide ($Li_{1.13}Ni_{0.39}Mn_{0.48}O_2$) tested in non-aqueous Li metal coin cell configuration enables a specific discharge capacity of 231.1 mAh g$^{-1}$, surpassing the 199.0 mAh g$^{-1}$ of the pristine positive electrode ($Li_{1.20}Ni_{0.36}Mn_{0.44}O_2$) at 20 mA g$^{-1}$, and also shows strongly suppressed voltage fade. Our work not only provides an understanding of honeycomb superstructures but also reveals the close relationship between superstructure units and oxygen redox activity. However, it is essential to acknowledge certain limitations of our study. While the modification with lower lithium content results in fewer $LiNiMn_5$ superstructure units, other parameters such as superstructure content, Li/Ni disordering, and even stacking faults also undergo changes. These parameters lie beyond the scope of the current study. Therefore, future research endeavors may focus on exploring these parameters and achieving a balance to enable high-energy-density, high-nickel Li-rich layered oxides.

## Methods

### Materials Synthesis

$Li_{1.2}Ni_xMn_{0.8-x}O_2$ ($x = 0.28, 0.32, 0.36, 0.40$) materials were synthesized using a hydroxide co-precipitation method followed by a solid-state reaction. First, a 2.0 M aqueous solution containing $NiSO_4·6H_2O$ ($\geq 98\%$, Sigma–Aldrich) and $MnSO_4·H_2O$ ($\geq 98\%$, Sigma–Aldrich) (molar ratios of 7:13, 2:3, 9:11 and 1:1 for N28, N32, N36 and N40, respectively), along with a 4.0 M NaOH solution ($\geq 98\%$, Sigma–Aldrich) and an aqueous 0.5 M $NH_3·H_2O$ solution (28%–30%, Sigma–Aldrich), were separately fed into a batch reactor with a volume of 1 L. The aqueous TM solution and $NH_3·H_2O$ solution were fed at the same rate while maintaining the pH value at $11.0 \pm 0.2$ by adjusting the feeding rate of the NaOH solution. The reaction process was carried out at a temperature of 50 °C and under a $N_2$ atmosphere. After a reaction duration of 20 h, followed by a 2-h aging process, the precursor was washed three times with deionized water and subsequently dried in an oven at 80 °C for 12 h under an air atmosphere. Finally, the target positive electrodes were obtained by mixing the precursor and $Li_2CO_3$ ($\geq 99\%$, Sigma–Aldrich) in a molar ratio of 1:1.5, followed by a heat treatment at 500 °C for 5 h and subsequently at 850 °C for 12 h under an air atmosphere. Similarly, other positive electrodes were prepared utilizing identical procedures except for a change in the ratio of $Li_2CO_3$ and precursor (1.2:1, 1.3:1, 1.4:1, and 1.5:1 for LM12, LM13, LM14, and LM15, respectively).

### Electrochemical measurements

The electrodes were prepared by thoroughly mixing the active material, Super C65 conductive carbon black (MTI Co., Ltd.), and polyvinylidene fluoride (PVDF, Sigma-Aldrich) in a mass ratio of 8:1:1 using a mortar. N-methyl-2-pyrrolidone (NMP) with a moisture content of less than 0.1% (VMR) was used as the solvent. The resulting mixture was then processed in a planetary mixer (THINKY ARV-310) with 2000 rpm for 10 min under an air atmosphere to ensure homogeneity. The resulting slurry was applied to aluminum foil (15 μm thickness, häberle LABORTECHNIK GmbH & Co.KG) at a speed of 25.0 mm s$^{-1}$ using a ZUA 2000 Universal applicator and the thickness was 150 nm. The coated aluminum foil was subsequently dried in an oven at 80 °C under ambient conditions to evaporate the NMP solvent. After a drying period of 6 h, the electrode material was punched into disks with a diameter of 12 mm (with an active material loading of approximately 3 mg cm$^{-2}$) using a handheld punch (NOGAMIGIKEN Co., Ltd.). A subsequent drying step was conducted using a Büchi glass oven (B-585) under vacuum conditions at 120 °C for 12 h. To evaluate the electrochemical performance, 2025-type coin cells were assembled. These cells comprised a Li metal anode (14 mm diameter, 0.25 mm thickness, PI-KEM), a single-layer Celgard 2500 membrane (25 μm thickness, 55% porosity, DODO Co., Ltd.) as the separator, and 80 μL of electrolyte consisting of 1 M lithium hexafluorophosphate (LiPF$_6$) dissolved in a solvent mixture of ethylene carbonate (EC), ethyl methyl carbonate (EMC), and dimethyl carbonate (DMC) in a volume ratio of 1:1:1, with moisture content maintained below 10 ppm (DODO Co., Ltd.). All assembly procedures were conducted within an argon-filled glovebox ($O_2$ and $H_2O$ < 0.1 ppm). The cells were cycled at a constant temperature of 25 °C using a Biologic VMP3 multichannel battery test system within a voltage range of 2.0–4.7 V. The charging protocol included a constant voltage step at 4.7 V for 10 min, followed by a 10-min resting period after each charge-discharge cycle. To ensure reproducibility, all electrochemical experiments were performed using a minimum of two-coin cells. Additionally, the current density for each cell was calculated based on the mass of the active material in the electrode. The coulombic efficiency was determined as the percentage ratio of discharging capacity to charging capacity, multiplied by 100. The average cell discharge voltage was defined as the voltage when the discharge capacity reached half of its maximum. For all ex-situ electrode measurements conducted in this work, which included SXAS, NMR, and

RIXS, the coin cells were subjected to specific voltage applications before being disassembled in the glovebox. The obtained electrodes were then thoroughly washed with DMC solvent three times and sealed under vacuum conditions within the glovebox.

## Materials characterizations

X-ray diffraction (XRD) measurements were measured by a Mo source powder diffractometer (STOE STADI P) in transmission geometry, with a wavelength of 0.7093 Å. The samples were loaded into capillaries with an outer diameter of 0.5 mm. The acquisition time for each pattern was 150 min. Synchrotron X-ray diffraction (SXRD) measurements were conducted at beamline P02.1, PETRA III (wavelength of -0.207 Å) at DESY in Hamburg[44,45]. The samples were loaded into capillaries with an outer diameter of 0.5 mm. Measured intensities were collected using a VAREX CT4343 detector (2880 × 2880 pixels, 150 × 150 μm² each) and the acquisition time of 120 s for each pattern. NIST SRM 660c (LaB₆) was used for geometry calibration performed with the software DAWN[46] followed by image integration including geometry, solid-angle, and polarization corrections. In-situ XRD measurements were conducted using an Ag source powder diffractometer (STOE STADI P, wavelength of 0.5594 Å) in transmission geometry. In-situ 2032-type coin cells with a Kapton window were employed. All cells were cycled at 20 mA g⁻¹ in a 2.0–4.7 V range for one cycle at 25 °C using a battery system (NOVA). The collection time for each pattern was set at 20 min. Rietveld refinements were carried out using the Fullprof software package[47].

Pair distribution function (PDF) measurements were performed at beamline ID31 at the European Synchrotron Radiation Facility (ESRF). The sample powders were loaded into cylindrical slots (approx. 1 mm thickness) held between Kapton windows in a high-throughput sample holder. Each sample was measured in a transmission with an incident X-ray energy of 75.00 keV (wavelength of 0.1653 Å). Measured intensities were collected using a Pilatus CdTe 2 M detector (1679 × 1475 pixels, 172 × 172 μm² each) positioned with the incident beam in the corner of the detector. The sample-to-detector distance was approximately 0.3 m for the total scattering measurement. Background measurements for the empty windows were measured and subtracted. NIST SRM 660b (LaB₆) was used for geometry calibration performed with the software pyFAI followed by image integration including a flat-field, geometry, solid-angle, and polarization corrections. The calculated PDF data with *R-3m* and *C2/m* was performed by PDFgui package[48].

The morphology of all samples was conducted by a scanning electron microscopy (SEM, Zeiss Merlin) with an acceleration voltage of 10 kV. The compositions of the samples were quantitatively determined by inductively coupled plasma-optical emission spectroscopy (ICP–OES) using a Thermo Fischer Scientific iCAP 7600 DUO. Scanning Transmission electron microscopy (STEM) energy dispersive X-ray spectroscopy (EDS) mapping and electron energy loss spectroscopy (EELS) experiments were conducted using a double aberration corrected Thermo-Fisher Themis-Z (operated at 300 kV equipped with a Super-X EDS detector and a Gaten GIF Continuum 970 HighRes EELS spectrometer). The structural information of samples was studied by high-resolution transmission electron microscopy (HRTEM). Each specimen was examined using Thermofisher Talos F200X TEM operated at 200 kV.

## Spectroscopy characterization

Magic-angle spinning (MAS) nuclear magnetic resonance (NMR) spectroscopy was performed on a Bruker Avance neo 200 MHz spectrometer with a magnetic field strength of 4.7 T. MAS spinning was carried out using 1.3 mm rotors at a frequency of 55 kHz. For the ⁶Li NMR experiments, ⁶LiOH·H₂O (95 atom%, Sigma–Aldrich) was utilized as the Li source for synthesis. The Larmor frequencies for ⁶Li and ⁷Li NMR were 29.5 MHz and 77.8 MHz, respectively. Spectra were acquired employing a rotor-synchronized Hahn-echo pulse sequence, with a 90° pulse length of 0.85 μs for ⁷Li and 1.6 μs for ⁶Li. All spectral shifts were referenced to an aqueous solution of LiCl (⁶LiCl for ⁶Li) at 0 ppm. The spectral intensities were normalized based on the sample mass and the number of scans. The NMR spectra were fitted using DMFIT program[49]. The atomic structures were plotted using VESTA software[50].

Hard X-ray absorption spectroscopy (XAS) experiments were performed at the XAS beamline of the KIT synchrotron in Karlsruhe. These measurements were conducted at 25 °C, employing the transmission mode for data acquisition. The obtained XAS data were processed using ATHENA software[51]. For soft X-ray absorption spectroscopy (SXAS), experiments were carried out at the WERA beamline at the KARA synchrotron in Karlsruhe. The Ni L-edge and O K-edge spectra were recorded using the fluorescence yield (FY) detection mode and the inverse partial fluorescence yield (iFY) for the Mn L-edge spectra. Resonant inelastic X-ray scattering (RIXS) experiments were conducted at beamline U41-PEAXIS at BESSY II, located at Helmholtz-Zentrum Berlin (HZB)[52]. A vacuum suitcase was used to transfer the sample from an N2-filled glovebox to the test chamber. The spectrometer was positioned at specular conditions relative to 60° scattering angle and was optimized to a combined resolution of 90 meV using a carbon tape. The O K-edge RIXS spectra of the samples were collected at an excitation energy of 531.0 eV. The acquisition time for each pattern was 30 min. RIXS mapping was measured in 0.5 eV energy steps from 526 eV to 534 eV. Data were processed using the Adler-4.0 software package.

## DFT calculations

The pre-optimization of the structure model was done by using the Vienna ab-initio simulation package (VASP) within the projector augmented-wave approach using the Perdew-Burke-Ernzerhof (PBE) with r²SCAN functional[53–57]. The correlation effects of the 3d-Ni and Mn orbitals (corresponding parameter $U_{Mn} = 1.8$ eV, $U_{Ni} = 2.1$ eV) and long-range dispersion with Grimme's D4 correction were taken into account[58,59]. A plane-wave energy cut-off of 450 eV and $2 \times 2 \times 2$ gamma centered *k*-point grid were used for the total energy.

To obtain reasonable electronic properties, the obtained structure model was reoptimized with CRYSTAL17 program version 1.0.2 using the PW1PW hybrid functional which has been successfully used, e.g., for the determination of transition metal oxide nitride phase stability and ortho phosphates before[60–62]. Long-range London dispersion was taken into account with Grimme's D3 correction with Becke-Johnson damping[63–65]. The empirical $s_8$ parameter was re-adjusted in preliminary calculations of the relative stability of $\alpha_{II}$-VOPO₄ and α-NbOPO₄, in which a modified value of $s_8 = 1.5363$ was obtained[66]. Several convergence parameters were changed to increase the numerical precision of the calculations. The truncation criteria for bielectronic integrals (TOLINTEG) were set to 7 7 7 7 14. To accelerate the SCF convergence, FMIXING was increased from default 30% to 85%. The Monkhorst-Pack shrinking factor is $4 \times 4$. The pob-TZVP-rev2 basis sets were used in all calculations[67,68]. Since Ni and Mn have magnetic moments, some pre-settings needed to be done. The calculation of the spins was done in an antiferromagnetic configuration with a maximum total spin of 66. Additionally, SPINLOCK was set to 66 30. The densities of states are calculated according to the Fourier-Legendre technique[69]. For a reliable result, the number of *k*-points must be set to 4 4.

The development of different structure models for Li₄₄Mn₁₈Ni₁₀O₇₂ (Li₁.₂₂Mn₀.₅₀Ni₀.₂₈O₂) was done by the program supercell[70]. The initial structure model is based on LiMnO₂ (S.G. *C2/m*, $a = 4.9370(10)$ Å, $b = 8.5320(10)$ Å, $c = 5.030(2)$ Å, $\beta = 109.46(3)°$). For the calculations the primitive cell was used ($a = 4.928$ Å, $b = 4.928$ Å, $c = 5.030$ Å, $\alpha = 80.35°$, $\beta = 99.60°$, $\gamma = 60.11°$) with adjusted occupancies for the metal cations Li, Mn and Ni (Supplementary Table 6).

For the given supercell input file with the partial S.O.F in the transition metal positions with a $3 \times 2 \times 2$ cell ca. $3.1 \bullet 10^8$ different combinations for $Li_{44}Mn_{18}Ni_{10}O_{72}$ were found. The structure model for $Li_{44}Mn_{18}Ni_{10}O_{72}$ was chosen based on low coulombic energy and the honeycomb superstructure $LiNiMn_5$. Supplementary Fig. 5a shows the exemplary crystal structure of the model. Atomic coordinates are given in Supplementary Table 7. The atomic structures were plotted using VESTA software.

## Data availability

Source data are provided with the paper. All data supporting the finding in the study are presented within the main text and the supplementary information.

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

## Acknowledgements

H. L. acknowledges the China Scholarship Council (CSC, No. 202106240017) scholarship. The authors gratefully acknowledge Valeriu Mereacre for helping with the material preparation and Liuda Mereacre for support in the lab. J.L. acknowledges the Fonds der Chemischen Industrie (FCI) for financial support. The authors would like to thank Bijian Deng for the SEM measurements. We acknowledge DESY (Hamburg, Germany), a member of the Helmholtz Association HGF, for the provision of experimental facilities. Parts of this research were carried out at PETRA III beamline P02.1 (proposal I-20221043). We are grateful to the KIT Light Source for the provision of beamtime and for support. We acknowledge the European Synchrotron Radiation Facility (ESRF) for provision of synchrotron radiation facilities. We would like to thank the Momentum Transfer team for facilitating the measurements and Jakub Drnec for assistance and support in using beamline ID31. The measurement setup was developed with funding from the European Union's Horizon 2020 research and innovation program under the STREAMLINE project (grant agreement ID 870313). The authors gratefully thank T. Bredow (University of Bonn) for valuable discussions. This research was supported in part by the high-performance computing resources provided by the Mulliken center (University of Bonn) and the HPC Keylab (University of Bayreuth).

## Author contributions

H.L. conceived the idea and conducted a discussion with S.I. while H.L. carried out the experiments. S.K. and M.B. performed DFT calculations. H.Li. performed the NMR measurements and data fitting. J.L. performed the PDF measurements and data analysis. O.D. and J.P. performed the in-situ XRD measurements. T.B. performed the ICP measurements. K.W., C.K., and W.H. performed the TEM measurements and data analysis. P.N., S.S., M.M., B.Y., and K.K. conducted the SXAS measurements and data analysis. S.M. performed the XAS measurements. D.W. performed the RIXS measurements and data analysis. V.B. performed the SXRD measurements. H.L. wrote the preliminary draft with input from W.H., M.K., H.E, and S.I. All authors contributed to interpreting the findings, reviewing, and revising the manuscript.

## Funding

## Competing interests

The authors declare no competing interests.

## Additional information

[1]Institute for Applied Materials (IAM), Karlsruhe Institute of Technology (KIT), Hermann-von-Helmholtz-Platz 1, Eggenstein-Leopoldshafen, Germany. [2]School of Chemical Engineering and Technology, Xi'an Jiaotong University, No.28, West Xianning Road, Xi'an, Shaanxi, China. [3]University of Bayreuth, Bavarian Center for Battery Technology (BayBatt), Universitätsstraße 30, Bayreuth, Germany. [4]School of Advanced Materials, Peking University, Shenzhen Graduate School, Shenzhen, China. [5]Institute of Nanotechnology (INT), Karlsruhe Institute of Technology (KIT), Hermann-von-Helmholtz-Platz 1, Eggenstein-Leopoldshafen, Germany. [6]Department of Materials and Earth Sciences, Technical University of Darmstadt, Darmstadt, Germany. [7]Helmholtz-Institute Ulm for Electrochemical Energy Storage (HIU), Karlsruhe Institute of Technology (KIT), Helmholtzstraße 11, Ulm, Germany. [8]Karlsruhe Nano Micro Facility, Karlsruhe Institute of Technology (KIT), Kaiserstraße 12, Karlsruhe, Germany. [9]Institute for Quantum Materials and Technologies, Karlsruhe Institute of Technology (KIT), Kaiserstraße 12, Karlsruhe, Germany. [10]Münster Electrochemical Energy Technology (MEET), University of Münster (WWU), Münster, Germany. [11]Institute for Photon Science and Synchrotron Radiation (IPS), Karlsruhe Institute of Technology (KIT), Hermann-von-Helmholtz-Platz 1, Eggenstein-Leopoldshafen, Germany. [12]Dynamics and Transport in Quantum Materials, Helmholtz-Zentrum Berlin für Materialen und Energie, GmbH, Albert-Einstein-Strasse 15, Berlin, Germany. [13]Deutsches Elektronen-Synchrotron (DESY), Notkestrasse 85, Hamburg, Germany. [14]Applied Chemistry and Engineering Research Centre of Excellence (ACER CoE), Université Mohammed VI Polytechnique (UM6P), Lot 660, Hay Moulay Rachid, Ben Guerir, Morocco. ✉e-mail: weibo.hua@xjtu.edu.cn; sylvio.indris@kit.edu

