## [Transparent Peer Review file · Nature Communications]

Tailoring superstructure units for improved oxygen redox activity in Li-rich layered oxide battery's positive electrodes

Corresponding Author: Dr Sylvio Indris

Version 0:

Reviewer comments:

Reviewer #1

(Remarks to the Author)

The authors revealed a direct correlation between oxygen redox activity and honeycomb superstructure units in Li-rich layered oxides, specifically the fractions of LiMn6 and Ni4+ stabilized LiNiMn5 within the TM layer and an excess of LiNiMn5 impedes the extraction/insertion of lithium ions during charging/discharging, resulting in incomplete oxygen redox activity. It is generally agreed upon that the increase in Ni content will reduce the specific capacity and the first cycle efficiency and result in an increase in Ni oxidation state from the reports from Qi et al (Electrochem 2023, 4, 21–30) and Chong et al (RSC Adv. 2016, 6, 53662). Here the authors show the similar phenomenon but with a reasonable explanation or mechanism for these conclusions, therefore I am unsure of the overall novelty - this should be addressed.

Overall, this manuscript could be published after a minor revision.

For Figure 2d and 2f, the signals are very weak, and the noise is higher, how much confidence you have for the data? Figure 2c, 2e and 2f, it seems that the authors only showed three points, where is the fourth point data/curve?

To address these intrinsic properties in high-nickel Li-rich layered oxides, the authors proposed to reduce the Li content during synthesis, please explain the rational design, why Li content determine the amount of the LiNiMn5? Also, why not just to optimize the Ni/Mn ratios instead?

Reviewer #2

(Remarks to the Author)

Li-rich layered oxides have high capacity because of high-voltage oxygen redox activity. Li-rich cathodes with a high nickel content exhibit improved voltage stability but suffer from poor discharge capacity. This work tailors the LiNiMn5 superstructure content by modification of Ni and Li contents. The study finds that an excess of LiNiMn5 impedes the extraction/insertion of lithium ions during charging/discharging, resulting in incomplete oxygen redox activity (retention of molecular O₂) at low potential (~3.3 V, versus Li⁺/Li). If the evidence is sufficient, the work is more interesting to open a new idea for enhanced oxygen redox of Li-rich layered oxides. There are several puzzling problems, so it is not recommended for publication:

1. In Li_{1.20}Ni_xMn_{0.8-x}O₂, x = 0.28, 0.32, 0.36, 0.40 materials, the content of Ni with +4 valence increases with the increase of x or the decrease of Li content. The work finds that the LiNiMn5 superstructure content increases with the increase of x or Li content. This is contradictory. The LiNiMn5 superstructure content is determined by the Li content or the ratio of Li/TM? Please give a reasonable interpretation.

2. In conventional LLOs such as Li_{1.2}Ni_{0.2}Mn_{0.6}O₂, generally, based on Pauling's electrostatic valence rule, LiMn6 configurations are preferably aggregated in Li₂MnO₃ due to their special stability. So it is difficult to permeate Ni to Li₂MnO₃. Many researches found that the breakdown of aggregated Li₂MnO₃, such as the permeation of TM with strong TM-O bonds, benefited the enhanced oxygen redox. Though the work gives evidence of LiNiMn5 configurations, why can the LiNiMn5 configurations exist in the work? Why is a reason that LiNiMn5 configurations result in incomplete oxygen redox activity?

3. The work does not provide TEM images, thus the other existence of Ni is unclear, and for example, the occupation of less Ni at Li-layer sites reduced oxygen release and lowered voltage decay.

4. The DOS of Li₄₄Mn₁₈Ni₁₀O₇₂ (Li_{1.22}Mn_{0.50}Ni_{0.28}O₂) is employed using DFT calculations. The state of Ni₄ is +4, as

expected in the work's model. Other models without the LiNiMn5 configurations are stable?
5. In addition, 231.1 mAh g⁻¹ at C/10 and a capacity retention of 84.8% after 200 cycles at C/3 is not high, such as 276.5 mAh g⁻¹ at C/10 and 85.4% after 300 cycles at 1 C of Li_{1.2}Ni_{0.2}Mn_{0.6}O₂(Li J, ACS nano, 2023, 17, 16827-16839).

Version 1:

Reviewer comments:

Reviewer #1

(Remarks to the Author)

After carefully reading the revised the manuscript, I have decided to approve the manuscript to proceed with the publication.

Thanks and let me know if you have any questions.

Best,

Gongshin

Reviewer #2

(Remarks to the Author)

The work is carefully modified. But there is one puzzling problem.

--The DOS of Li₄₄Mn₁₈Ni₁₀O₇₂ (Li_{1.22}Mn_{0.50}Ni_{0.28}O₂) is employed using DFT calculations. The state of Ni₄ is +4, as expected in the work's model. So the structural stability with one LiNiMn5 configuration instead of the state of Ni₄ with +4 should be explored, because the chemical valence of Ni₄ is related to the structure.

In addition, to improve the redox reversibility of lattice oxygen and cycling performance, recently two works (ACS Nano 2023, 17, 16827–16839; Nano Energy, 2024, 123, 109390) are done though tuning the Li₂MnO₃ domain, please cite them.

Version 2:

Reviewer comments:

Reviewer #2

(Remarks to the Author)

it is recommended for publication.

REVIEWER COMMENTS

Reviewer #1 (Remarks to the Author):

The authors revealed a direct correlation between oxygen redox activity and honeycomb superstructure units in Li-rich layered oxides, specifically the fractions of LiMn_6 and Ni^{4+} stabilized LiNiMn_5 within the TM layer and an excess of LiNiMn_5 impedes the extraction/insertion of lithium ions during charging/discharging, resulting in incomplete oxygen redox activity. It is generally agreed upon that the increase in Ni content will reduce the specific capacity and the first cycle efficiency and result in an increase in Ni oxidation state from the reports from Qi et al (*Electrochem* **2023**, 4, 21–30) and Chong et al (*RSC Adv.* **2016**, 6, 53662). Here the authors show the similar phenomenon but with a reasonable explanation or mechanism for these conclusions, therefore I am unsure of the overall novelty - this should be addressed.

Overall, this manuscript could be published after a minor revision.

For Figure 2d and 2f, the signals are very weak, and the noise is higher, how much confidence you have for the data? Figure 2c, 2e and 2f, it seems that the authors only showed three points, where is the fourth point data/curve?

To address these intrinsic properties in high-nickel Li-rich layered oxides, the authors proposed to reduce the Li content during synthesis, please explain the rational design, why Li content determine the amount of the LiNiMn_5 ? Also, why not just to optimize the Ni/Mn ratios instead?

Reviewer #2 (Remarks to the Author):

Li-rich layered oxides have high capacity because of high-voltage oxygen redox activity. Li-rich cathodes with a high nickel content exhibit improved voltage stability but suffer from poor discharge capacity. This work tailors the LiNiMn_5 superstructure content by modification of Ni and Li contents. The study finds that an excess of LiNiMn_5 impedes the extraction/insertion of lithium ions during charging/discharging, resulting in incomplete oxygen redox activity (retention of molecular O_2) at low potential (~ 3.3 V, versus Li^+/Li). If the evidence is sufficient, the work is more interesting to open a new idea for enhanced oxygen redox of Li-rich layered oxides. There are several puzzling problems, so it is not recommended for publication:

1. In $\text{Li}_{1.20}\text{Ni}_x\text{Mn}_{0.8-x}\text{O}_2$, $x = 0.28, 0.32, 0.36, 0.40$ materials, the content of Ni with +4 valence increases with the increase of x or the decrease of Li content. The work finds that the LiNiMn_5 superstructure content increases with the increase of x or Li content. This is contradictory. The LiNiMn_5 superstructure content is determined by the Li content or the ratio of Li/TM? Please give a reasonable interpretation.

2. In conventional LLOs such as $\text{Li}_{1.2}\text{Ni}_{0.2}\text{Mn}_{0.6}\text{O}_2$, generally, based on Pauling's electrostatic valence rule, LiMn_6 configurations are preferably aggregated in Li_2MnO_3 due to their special stability. So it is difficult to permeate Ni to Li_2MnO_3 . Many researches found that the breakdown of aggregated Li_2MnO_3 , such as the permeation of TM with strong TM-O bonds, benefited the enhanced oxygen redox. Though the work gives evidence of LiNiMn_5 configurations, why can the LiNiMn_5 configurations exist in the work? Why is a reason that LiNiMn_5 configurations result in incomplete oxygen redox activity?
3. The work does not provide TEM images, thus the other existence of Ni is unclear, and for example, the occupation of less Ni at Li-layer sites reduced oxygen release and lowered voltage decay.
4. The DOS of $\text{Li}_{44}\text{Mn}_{18}\text{Ni}_{10}\text{O}_{72}$ ($\text{Li}_{1.22}\text{Mn}_{0.50}\text{Ni}_{0.28}\text{O}_2$) is employed using DFT calculations. The state of Ni4 is +4, as expected in the work's model. Other models without the LiNiMn_5 configurations are stable?
5. In addition, 231.1 mAh g^{-1} at C/10 and a capacity retention of 84.8% after 200 cycles at C/3 is not high, such as 276.5 mAh g^{-1} at C/10 and 85.4% after 300 cycles at 1 C of $\text{Li}_{1.2}\text{Ni}_{0.2}\text{Mn}_{0.6}\text{O}_2$ (Li J, *ACS Nano*, **2023**, 17, 16827-16839).

Point-to-point Response to Reviewers' Comments

Tailoring Superstructure Units for Enhanced Oxygen Redox in Li-Rich Layered Oxides

(Manuscript ID NCOMMS-24-16929)

We sincerely thank the editor and all reviewers for their valuable review comments that we have used to improve the quality of our manuscript. The reviewer comments are laid out below and specific concern have been numbered, and all the responses are listed in the following section, point by point.

Reviewer #1 (Remarks to the Author):

The authors revealed a direct correlation between oxygen redox activity and honeycomb superstructure units in Li-rich layered oxides, specifically the fractions of LiMn_6 and Ni^{4+} stabilized LiNiMn_5 within the TM layer and an excess of LiNiMn_5 impedes the extraction/insertion of lithium ions during charging/discharging, resulting in incomplete oxygen redox activity. It is generally agreed upon that the increase in Ni content will reduce the specific capacity and the first cycle efficiency and result in an increase in Ni oxidation state from the reports from Qi et al (*Electrochem* **2023**, 4, 21–30) and Chong et al (*RSC Adv.* **2016**, 6, 53662). Here the authors show the similar phenomenon but with a reasonable explanation or mechanism for these conclusions, therefore I am unsure of the overall novelty - this should be addressed.

Overall, this manuscript could be published after a minor revision.

Response: Thank you for the high evaluation of our work. In the following, we will address all the comments sequentially.

1. It is generally agreed upon that the increase in Ni content will reduce the specific capacity and the first cycle efficiency and result in an increase in Ni oxidation state from the reports from Qi et al (*Electrochem* **2023**, 4, 21–30) and Chong et al (*RSC Adv.* **2016**, 6, 53662). Here the authors show the similar phenomenon but with a reasonable explanation or mechanism for these conclusions, therefore I am unsure of the overall novelty - this should be addressed.

Response: We appreciate the reviewer's comment. As indicated by previous publications (*Electrochem* **2023**, 4, 21–30; *RSC Adv.* **2016**, 6, 53662; *J. Mater. Chem. A.* **2015**, 3, 22199), there has been considerable interest in elucidating the role of nickel content in the electrochemistry of Li-rich layered oxides. Unfortunately, these studies primarily focused on

electrochemical characterization and did not provide a satisfactory explanation for the poor electrochemical performance of high-nickel Li-rich layered oxides. Moreover, no universal solutions to address this issue have been identified. Our research fills this gap by offering a reasonable explanation and potential solutions to improve the electrochemistry of high-nickel Li-rich layered oxides.

Based on earlier studies, our work innovatively employed a series of advanced techniques, including SXR, PDF, SXAS, RIXS, TEM (addressing Question 3 from Reviewer #2), NMR, DFT calculations, and electrochemical investigations. This comprehensive approach provides definitive answers to critical questions. Firstly, we identified the types of superstructure units within the transition metal layer and, for the first time, elucidate the influence of the nickel ions' valence state on these units. More importantly, we successfully revealed the intrinsic correlation between oxygen redox activity and superstructure units, elucidating the mechanism behind the poor discharge capacity observed in high-nickel Li-rich cathodes. Additionally, beyond elucidating the mechanism, we proposed a potential solution to tailor superstructure units to enhance battery performance with reversible oxygen redox.

Therefore, we believe that our conclusive evidence on tailoring superstructure units for enhanced oxygen redox in Li-rich layered oxides will be of significant interest to both the battery community and the broader readership of Nature Communications.

Changes made:

In the revision manuscript, we inserted additional references in the introduction on page 3.

20. Qi, G. et al. Impact of Ni Content on the Electrochemical Performance of the Co-Free, Li and Mn-Rich Layered Cathode Materials. *Electrochem* 4, 21–30 (2023).

21. Knight, J. C. & Manthiram, A. Effect of nickel oxidation state on the structural and electrochemical characteristics of lithium-rich layered oxide cathodes. *J. Mater. Chem. A* 3, 22199–22207 (2015).

2. For Figure 2d and 2f, the signals are very weak, and the noise is higher, how much confidence you have for the data? Figure 2c, 2e and 2f, it seems that the authors only showed three points, where is the fourth point data/curve?

Response:

We appreciate the reviewer's comments and fully understand their concerns regarding the data quality in the manuscript. In Fig. 2, our primary aim is to identify the redox nature of the dQ/dV peak observed below 3.7 V and elucidate the mechanism contributing to the poor

capacity in high-nickel Li-rich cathodes. As discussed in the manuscript, this peak is typically associated with the reduction of oxygen (O) or manganese (Mn). Therefore, understanding the redox evolution of O and Mn, represented by changes in the O K-edge and Mn L-edge spectra, is crucial during charge and discharge.

In our study, we selected four points—pristine, charged to 4.45 V, charged to 4.7 V, and discharged to 2.0 V—during the first cycle to collect the Ni, Mn L-edge, and O K-edge spectra with soft-XAS spectroscopy (Fig. 2a-d). The spectra of the Mn L-edge significantly overlap in Fig. 2c. To enhance clarity, we present the stack of Mn L-edge spectra in Fig. R2. Combining these with the O K-edge SXAS results, we identify that the incomplete oxygen redox occurs at a low potential (3.3 V vs. Li^+/Li) in Li-rich layered oxides for a high-nickel system.

Given this abnormal phenomenon, we additionally conducted O K-edge RIXS measurements (Fig. 2e,f) to further validate our findings. As depicted in Fig. 2e-f, a distinct characteristic peak C (around 7.8 eV) and a vibrational peak D around the elastic region (0 eV) are still observed at discharge to 2.0 V. This finding is consistent with previous O K-edge SXAS results, further supporting incomplete oxygen redox. Our data and findings are reasonable and well-justified. Additionally, similar O K-edge RIXS signals are documented in previous studies (*Nat Commun.* **2021**, 12, 3071; *Nat Commun.* **2022**, 13, 1123; *J. Am. Chem. Soc.* **2023**, 18, 10208-10219; *Nat Energy.* **2024**, 9, 184–196).

The absence of data at 4.45 V for the RIXS measurements does not influence the conclusion of this work. Therefore, we selected three of the most critical states for RIXS measurements (excluding the point at 4.45 V) to address our research questions. We hope for the reviewer's understanding.

Fig. R2. SXAS results of the Mn L-edge collected at pristine, charge to 4.45 V, charge to 4.7 V, and discharge to 2.0 V states.

3. To address these intrinsic properties in high-nickel Li-rich layered oxides, the authors proposed to reduce the Li content during synthesis, please explain the rational design, why Li content determine the amount of the LiNiMn₅? Also, why not just to optimize the Ni/Mn ratios instead?

Response: We appreciate the reviewer for the careful review on our submitted work. Regarding why Li content determine the amount of the LiNiMn₅, our findings reveal that the LiNiMn₅ superstructure units are exclusively stabilized by Ni⁴⁺ and not by Ni²⁺ or Ni³⁺. Therefore, to reduce the presence of LiNiMn₅ units, the overall Ni valence state in Li-rich cathodes must be decreased. In our work, reducing the lithium content during synthesis effectively increases the overall transition metal (TM) content in the TM layer, as demonstrated by the increase from 0.80 in LM15 (Li_{1.2}Ni_{0.36}Mn_{0.44}O₂) to 0.87 in LM13 (Li_{1.13}Ni_{0.39}Mn_{0.48}O₂). XAS results confirm the decrease in the valence state of Ni, leading to a corresponding change in superstructure units. Specifically, the LiNiMn₅/(LiNiMn₅ + LiMn₆) peak ratio decreases to 42.7% for LM13, compared to 58.2% for LM15, as evidenced by NMR. As described in the response to Question 1 from reviewer #2, for a given Ni/Mn ratio, the LiNiMn₅ superstructure content is determined by the Li content. This is because the Ni valence state changes as the Li content changes.

Regarding, why not just to optimize the Ni/Mn ratios instead, we thank the reviewer for pointing out this method for tailoring LiNiMn₅ units. We fully acknowledge the reviewer's suggestion to tailor LiNiMn₅ units by further optimizing the Ni/Mn ratios in Li-rich cathodes. As evident from the electrochemical data (Fig. 1c-d) and the changes in superstructure units (Fig. 3a) observed in these cathodes with varying nickel contents, there may indeed be an optimal Ni/Mn ratio that can achieve both high capacity and high voltage.

However, our primary aim is to elucidate the underlying mechanism behind the unsatisfactory electrochemistry observed in high-nickel Li-rich layered oxides. Therefore, we have chosen a practical and simple approach to validate the accuracy of our theoretical findings. Additionally, implementing the method of optimizing Ni/Mn ratios may require a significant number of precursor synthesis experiments, which falls beyond the scope of the current manuscript. Moreover, adjusting the lithium content provides additional benefits, such as modifying the superstructure content to fine-tune cation/anion redox contributions. This adjustment promotes higher cation redox, thereby enhancing structural stability compared to the instability

associated with anion redox at high voltage. Moreover, moderate Li/Ni disordering promotes structural reversibility due to the pillar effect. Finally, this approach represents a practical and simple solution for industry applications when compared to other complex modification processes. Considering all these factors, the rationale for reducing Li content is reasonable and easily justified.

Reviewer #2 (Remarks to the Author):

Li-rich layered oxides have high capacity because of high-voltage oxygen redox activity. Li-rich cathodes with a high nickel content exhibit improved voltage stability but suffer from poor discharge capacity. This work tailors the LiNiMn₅ superstructure content by modification of Ni and Li contents. The study finds that an excess of LiNiMn₅ impedes the extraction/insertion of lithium ions during charging/discharging, resulting in incomplete oxygen redox activity (retention of molecular O₂) at low potential (~3.3 V, versus Li⁺/Li). If the evidence is sufficient, the work is more interesting to open a new idea for enhanced oxygen redox of Li-rich layered oxides. There are several puzzling problems, so it is not recommended for publication:

Response: Thank you for the careful and in-depth review of our manuscript. In the following, we will address all the comments and revise the manuscript accordingly.

1. In Li_{1.20}Ni_xMn_{0.8-x}O₂, x = 0.28, 0.32, 0.36, 0.40 materials, the content of Ni with +4 valence increases with the increase of x or the decrease of Li content. The work finds that the LiNiMn₅ superstructure content increases with the increase of x or Li content. This is contradictory. The LiNiMn₅ superstructure content is determined by the Li content or the ratio of Li/TM? Please give a reasonable interpretation.

Response: We appreciate the reviewer's comment. Our findings reveal that the LiNiMn₅ superstructure units are exclusively stabilized by Ni⁴⁺, rather than by Ni²⁺ or Ni³⁺. This stabilization is facilitated by the presence of Ni⁴⁺, which promotes the formation of LiNiMn₅ units in Li-rich cathodes.

In materials such as Li_{1.20}Ni_xMn_{0.8-x}O₂ (x = 0.28, 0.32, 0.36, 0.40), with a Li/TM ratio of 1.5, an increase in x results in an increase in the valence state of Ni. As a consequence, Li_{1.2}Ni_{0.40}Mn_{0.40}O₂ (N40), with a higher Ni valence state, contains a greater proportion of LiNiMn₅ units compared to Li_{1.2}Ni_{0.28}Mn_{0.52}O₂ (N28).

Furthermore, for a given Ni/Mn ratio, reducing the lithium content during synthesis effectively increases the overall transition metal (TM) content in the TM layer, as demonstrated by the increase from 0.80 in LM15 (Li_{1.2}Ni_{0.36}Mn_{0.44}O₂) to 0.87 in LM13 (Li_{1.13}Ni_{0.39}Mn_{0.48}O₂). As a result, there are fewer LiNiMn₅ units in the LM13 with lower valence state of Ni.

Therefore, the LiNiMn₅ superstructure content in Li-rich cathodes is not simply determined by the Li content or the ratio of Li/TM; rather, it changes only when the Ni valence state changes.

2. In conventional LLOs such as $\text{Li}_{1.2}\text{Ni}_{0.2}\text{Mn}_{0.6}\text{O}_2$, generally, based on Pauling's electrostatic valence rule, LiMn_6 configurations are preferably aggregated in Li_2MnO_3 due to their special stability. So it is difficult to permeate Ni to Li_2MnO_3 . Many researches found that the breakdown of aggregated Li_2MnO_3 , such as the permeation of TM with strong TM-O bonds, benefited the enhanced oxygen redox. Though the work gives evidence of LiNiMn_5 configurations, why can the LiNiMn_5 configurations exist in the work? Why is a reason that LiNiMn_5 configurations result in incomplete oxygen redox activity?

Response: We are grateful to the reviewer for pointing out Pauling's electrostatic valence principle as a demonstration of the stability of the superstructure configuration. Indeed, we fully agree with the reviewer that the LiMn_6 units are more likely to form in Li_2MnO_3 compared to other superstructure units. Furthermore, previous studies (*Matter.* **2022**, 5, 3869-3882; *Materials Today.* **2021**, 51, 15-26) also suggest that the Ni is difficult to detect within Li_2MnO_3 -like domains. However, the above conclusion is only based on the case in which Ni has a valence state of 2 or 3. The possibility of the presence of Ni^{4+} in LiNiMn_5 superstructure units has been previously overlooked. Note that while Ni^{4+} is atypical in conventional layered cathodes, its presence in Li-rich cathode systems is plausible due to the lithium excess structural configuration (*Chem. Mater.* **2020**, 32, 9211–9227).

Based on Pauling's electrostatic valence principle (*J. Am. Chem. Soc.* **1929**, 51, 1010-1026) we can apply a semiquantitative equation to evaluate the stability of a coordination configuration (*Materials Today.* **2021**, 51, 15-26):

$$\Delta Z = \left| Z_A - \sum \frac{Z_c}{n} \right|$$

According to the principle of local electrical neutrality, a structure may become unstable if $\Delta Z > 0$. In Li-rich cathodes, three Li ions in the Li layer are consistently coordinated with central oxygen atom, while the coordination environments of cations in the transition metal layer may have different combinations. The most common LiMn_6 units, equivalent to $\text{Li}_3\text{-O-LiMn}_2$, exhibit stable state with $\Delta Z = 0$. In our work, LiNiMn_5 units ($\text{Li}_3\text{-O-LiNiMn}$) also demonstrate $\Delta Z = 0$, attributed to the presence of Ni^{4+} rather than Ni^{2+} ($\Delta Z = 1/3$) and Ni^{3+} ($\Delta Z = 1/6$). Moreover, our ^6Li NMR results clearly demonstrate the presence of these LiNiMn_5 units in these Li-rich cathodes.

Regarding Why the LiNiMn_5 configurations result in incomplete oxygen redox activity, previous theoretical calculations (*Matter.* **2022**, 5, 3869-3882) indicate that the Li_4MnNi -coordinated O, referred to as LiNiMn_5 in our work, has a much lower cationic and anionic redox activity compared to LiMn_6 units, suggesting a poor electrochemical activity. Additionally, Li moves from one octahedral site to another octahedral site by passing through an intermediate

tetrahedral site where it encounters strong repulsion from a nearby transition metal. The associated activation barriers for Li movement with Ni⁴⁺, Ni³⁺ and Ni²⁺ are 490, 310, and 210 meV, respectively (*Science*. **2006**, 311, 977-980). Therefore, these theoretical findings suggest that Ni⁴⁺-stabilized LiNiMn₅ may impede the movement of lithium ions during charging/discharging. Moreover, Li-rich cathodes undergo a cationic–anionic redox inversion with significant chemical and electrochemical asymmetry between charge and discharge (*Nature Mater.* **2022**, 21, 1165-1174). Consequently, at low potential during discharge, some Li ions fail to return to the transition metal layer to coordinate with O, resulting in incomplete oxygen redox. This observation is further supported by our experimental results, including SXAS, RIXS, ex-situ NMR, in-situ XRD, and electrochemistry.

Changes made:

In the revised manuscript, we have included additional content on page 10: “In addition, Pauling's electrostatic valence principle³³ and semi-quantitative equation³⁴ can be used to evaluate the possibility of a coordination configuration. According to principle of local electrical neutrality, a structure may become unstable if the local difference in valences of anions/cations $\Delta Z > 0$. The most common LiMn₆ units, equivalent to Li₃-O-LiMn₂, exhibit a stable state with $\Delta Z = 0$. Similarly, LiNiMn₅ units (Li₃-O-LiNiMn) also demonstrate $\Delta Z = 0$, attributed to the presence of Ni⁴⁺ rather than Ni²⁺ ($\Delta Z = 1/3$) and Ni³⁺ ($\Delta Z = 1/6$). This elucidates why prior investigations failed to detect nickel within Li₂MnO₃-like domains, given that the possibility of the presence of Ni⁴⁺ in LiNiMn₅ superstructure units has been previously overlooked^{34,35}. Previous theoretical calculations³⁵ indicate that the Li₄MnNi-coordinated O, referred to as LiNiMn₅ in our work, exhibits significantly lower cationic and anionic redox activity compared to LiMn₆ units, indicating limited electrochemical activity. Moreover, lithium ions move from one octahedral site to another one, passing through an intermediate tetrahedral site where they encounter strong repulsion from nearby transition metals. The associated activation barriers for lithium movement with Ni⁴⁺, Ni³⁺ and Ni²⁺ have been calculated to be 490, 310, and 210 meV, respectively³⁶. These theoretical findings imply that LiNiMn₅ may impede the extraction/insertion of lithium ions. During charging/discharging, Li-rich cathodes experience a cationic–anionic redox inversion, leading to electrochemical asymmetry²³. As a consequence, at low discharge potentials, some lithium ions fail to return to the transition metal layer to coordinate with O, resulting in incomplete oxygen redox. This analysis is consistent with the findings from our O K-edge SXAS and RIXS experiments.”

23. Li, B. et al. Capturing dynamic ligand-to-metal charge transfer with a long-lived cationic intermediate for anionic redox. *Nat. Mater.* 21, 1370–1379 (2022).

33. Pauling, L. The principles determining the structure of complex ionic crystals. *J. Am. Chem. Soc.* **51**, 1010–1026 (1929).
34. Yin, C. *et al.* Structural insights into composition design of Li-rich layered cathode materials for high-energy rechargeable battery. *Materials Today* **51**, 15–26 (2021).
35. Yang, Y. *et al.* Cation configuration in transition-metal layered oxides. *Matter* **5**, 3869–3882 (2022).
36. Kang, K., Meng, Y. S., Bréger, J., Grey, C. P. & Ceder, G. Electrodes with High Power and High Capacity for Rechargeable Lithium Batteries. *Science* **311**, 977–980 (2006).

3. The work does not provide TEM images, thus the other existence of Ni is unclear, and for example, the occupation of less Ni at Li-layer sites reduced oxygen release and lowered voltage decay.

Response: We appreciate the reviewer's comment and suggestion. To investigate the potential influence of cation mixing (where Ni occupies Li sites) on the electrochemistry of Li-rich cathodes with varying nickel contents, we conducted transmission electron microscopy (TEM) measurements. The high-resolution TEM (HRTEM) images of N28, N32, N36, and N40, as depicted in Fig. R3, reveal clear lattice fringes without obvious evidence of cation mixing. Moreover, the observed average interplanar spacing of approximately 0.47 nm corresponds to the (003)_R or (001)_M plane of the layered rhombohedral (*R-3m*) or monoclinic (*C2/m*) structure. Additionally, integrating the previously conducted high-resolution synchrotron X-ray diffraction (SXR) measurements, we conclude that cation mixing has little impact on the electrochemistry of these cathodes. This further underscores the close correlation between honeycomb superstructure units and electrochemistry in our work.

Changes made:

In the revised manuscript, we have included additional content on page 6 in the manuscript and page 5 in the supporting information “The high-resolution transmission electron microscopy (HRTEM) images of N28, N32, N36, and N40, as shown in Supplementary Fig. 4, reveal clear lattice fringes without obvious evidence of cation mixing. Moreover, the observed average interplanar spacing of approximately 0.47 nm corresponds to the (003)_R or (001)_M plane of the layered rhombohedral (*R-3m*) or monoclinic (*C2/m*) structure.”

Additionally, we have included additional content on the page 20 in the Methods section “Structural information about the samples was obtained by high-resolution transmission electron microscopy (HRTEM). Each specimen was examined using Thermofisher Talos F200X TEM operated at 200 kV.”

Fig. R3. HRTEM images of N28 (a-b), N32 (c-d), N36 (e-f), and N40 (g-h) powder.

4. The DOS of $\text{Li}_{44}\text{Mn}_{18}\text{Ni}_{10}\text{O}_{72}$ ($\text{Li}_{1.22}\text{Mn}_{0.50}\text{Ni}_{0.28}\text{O}_2$) is employed using DFT calculations. The state of Ni4 is +4, as expected in the work's model. Other models without the LiNiMn_5 configurations are stable?

Response: We appreciate the reviewer's comment. We also performed calculations on a model without the LiNiMn_5 unit of $\text{Li}_{44}\text{Mn}_{18}\text{Ni}_{10}\text{O}_{72}$ ($\text{Li}_{1.22}\text{Mn}_{0.50}\text{Ni}_{0.28}\text{O}_2$). The energy difference between this model and the one containing the LiNiMn_5 unit was minimal, approximately 0.005

eV/atom. As a result, it can be concluded that their energies are the same, and they can be stable.

5. In addition, 231.1 mAh g⁻¹ at C/10 and a capacity retention of 84.8% after 200 cycles at C/3 is not high, such as 276.5 mAh g⁻¹ at C/10 and 85.4% after 300 cycles at 1 C of Li_{1.2}Ni_{0.2}Mn_{0.6}O₂ (Li J, *ACS Nano*, **2023**, 17, 16827-16839).

Response: We appreciate the reviewer for the careful review on our submitted work. We fully agree with the reviewer that our cathode (LM13) exhibits lower discharge capacity compared to the classical Li-rich cathode (Li_{1.2}Ni_{0.2}Mn_{0.6}O₂). Despite achieving high capacity and stability by various modification in recent years, these Li-rich cathodes still encounter substantial voltage decay during cycling. As highlighted in the introduction, optimizing the chemical composition has emerged as a promising approach to mitigate voltage decay in layered Li-rich cathodes (*Adv. Energy Mater.* **2018**, 8, 1800606; *ACS Appl. Mater. Interfaces.* **2016**, 8, 20138–20146). Additionally, high-nickel Li-rich layered oxides offer several advantages in stabilizing voltage and reducing oxygen release (*Nat. Mater.* **2023**, 22, 1370–1379). However, high-nickel Li-rich cathodes, such as Li_{1.2}Ni_{0.4}Mn_{0.4}O₂, have garnered less attention due to their lower discharge capacity (less than 200 mAh g⁻¹ at C/10) compared to low-nickel Li-rich cathodes (*Electrochem* **2023**, 4, 21–30; *RSC Adv.* **2016**, 6, 53662; *J. Mater. Chem. A.* **2015**, 3, 22199). Thus, the key to advancing Li-rich cathodes is achieving voltage stability in high-nickel systems without compromising discharge capacity.

Therefore, the primary aim of this work is to elucidate the underlying mechanism behind the unsatisfactory electrochemistry observed in high-nickel Li-rich layered oxides. Building on the insights gained from the mechanism, we propose a practical and straightforward solution to tailor superstructure units, thereby enhancing oxygen redox reversibility and improving battery performance (from 199.0 to 231.1 mAh g⁻¹ at C/10). Our strategy has proven to be efficient compared to pristine high-nickel Li-rich cathodes. As described in the discussion in the manuscript, adjusting the lithium content during synthesis, in addition to changes in superstructure units, affects other parameters such as superstructure content, Li/Ni disorder, and stacking faults. Therefore, future research endeavours may focus on exploring these parameters and achieving a balance to further enable high-energy-density, high-nickel Li-rich layered oxides.

REVIEWER COMMENTS

Reviewer #1 (Remarks to the Author):

After carefully reading the revised the manuscript, I have decided to approve the manuscript to proceed with the publication.

Thanks and let me know if you have any questions.

Best,

Gongshin

Reviewer #2 (Remarks to the Author):

The work is carefully modified. But there is one puzzling problem.

--The DOS of $\text{Li}_{44}\text{Mn}_{18}\text{Ni}_{10}\text{O}_{72}$ ($\text{Li}_{1.22}\text{Mn}_{0.50}\text{Ni}_{0.28}\text{O}_2$) is employed using DFT calculations. The state of Ni4 is +4, as expected in the work's model. So the structural stability with one LiNiMn_5 configuration instead of the state of Ni4 with +4 should be explored, because the chemical valence of Ni4 is related to the structure.

In addition, to improve the redox reversibility of lattice oxygen and cycling performance, recently two works (*ACS Nano* **2023**, 17, 16827–16839; *Nano Energy*, **2024**, 123, 109390) are done though tuning the Li_2MnO_3 domain, please cite them.

Point-to-point Response to Reviewers' Comments

Tailoring Superstructure Units for Enhanced Oxygen Redox in Li-Rich Layered Oxides

(Manuscript ID NCOMMS-24-16929A)

We sincerely thank the editor and all reviewers for their valuable review comments that we have used to improve the quality of our manuscript. The reviewer comments are laid out below and specific concern have been numbered, and all the responses are listed in the following section, point by point.

Reviewer #1 (Remarks to the Author):

After carefully reading the revised the manuscript, I have decided to approve the manuscript to proceed with the publication.

Thanks and let me know if you have any questions.

Best,

Gongshin

Response: We highly appreciate the reviewer's positive comments.

Reviewer #2 (Remarks to the Author):

The work is carefully modified. But there is one puzzling problem.

--The DOS of $\text{Li}_{44}\text{Mn}_{18}\text{Ni}_{10}\text{O}_{72}$ ($\text{Li}_{1.22}\text{Mn}_{0.50}\text{Ni}_{0.28}\text{O}_2$) is employed using DFT calculations. The state of Ni4 is +4, as expected in the work's model. So the structural stability with one LiNiMn_5 configuration instead of the state of Ni4 with +4 should be explored, because the chemical valence of Ni4 is related to the structure.

In addition, to improve the redox reversibility of lattice oxygen and cycling performance, recently two works (*ACS Nano* **2023**, 17, 16827–16839; *Nano Energy*, **2024**, 123, 109390) are done though tuning the Li_2MnO_3 domain, please cite them.

Response: Thank you for the careful and in-depth review of our manuscript. In the following, we will address all the comments and revise the manuscript accordingly.

1. The DOS of $\text{Li}_{44}\text{Mn}_{18}\text{Ni}_{10}\text{O}_{72}$ ($\text{Li}_{1.22}\text{Mn}_{0.50}\text{Ni}_{0.28}\text{O}_2$) is employed using DFT calculations. The state of Ni4 is +4, as expected in the work's model. So the structural stability with one LiNiMn_5 configuration instead of the state of Ni4 with +4 should be explored, because the chemical valence of Ni4 is related to the structure.

Response: We appreciate the reviewer's comment. We have calculated the oxidation state of Ni in the LiNiMn_5 configuration without a priori assumptions on the Ni oxidation state. The relaxed structure shows that Ni possesses the oxidation state +4. In addition, we have calculated models with the same overall composition but without the LiNiMn_5 configuration (always without assuming the oxidation state a priori), in which we find that Ni has an oxidation state lower than +4. Based on that we conclude the LiNiMn_5 configuration stabilize Ni in the oxidation state +4 in the structure.

2. In addition, to improve the redox reversibility of lattice oxygen and cycling performance, recently two works (*ACS Nano* **2023**, 17, 16827–16839; *Nano Energy*, **2024**, 123, 109390) are done though tuning the Li_2MnO_3 domain, please cite them.

Response: We are grateful to the reviewer for pointing out the importance of the tuning of Li_2MnO_3 domain size or distribution to the oxygen redox reversibility of Li-rich cathodes. To better reflect this recent progress, we have added the relevant content in the Introduction and cited the relevant studies.

Changes made:

In the revised manuscript, we have included additional content on page 3: “For instance, a consensus has emerged from relevant studies, highlighting the detrimental role of localized superstructure domains characterized by in-plane Li/Mn order of Li-rich cathodes, while delocalized or dispersed domains characterized by in-plane Li/Mn disorder effectively improve the oxygen redox reversibility and voltage stability^{10–12}.”

10. Li, J. et al. Tuning Li_2MnO_3 -Like Domain Size and Surface Structure Enables Highly-stabilized Li-Rich Layered Oxide Cathodes. *ACS Nano* **17**, 16827–16839 (2023).

11. Guo, X. et al. Weak σ – π – σ interaction stabilizes oxygen redox towards high-performance Li-rich layered oxide cathodes. *Nano Energy* **123**, 109390 (2024).

12. Zhang, M. et al. Formulating Local Environment of Oxygen Mitigates Voltage Hysteresis in Li-Rich Materials. *Advanced Materials* **36**, 2311814 (2024).